# Lunar farside volcanism 2.8 billion years ago from Chang'e-6 basalts

Qian W. L. Zhang[1], Mu-Han Yang[1,2], Qiu-Li Li[1,2 ✉], Yu Liu[1], Zong-Yu Yue[1], Qin Zhou[3], Liu-Yang Chen[1,2], Hong-Xia Ma[1], Sai-Hong Yang[3], Xu Tang[1], Guang-Liang Zhang[3], Xin Ren[3] & Xian-Hua Li[1,2]

Unravelling the volcanic history of the enigmatic lunar farside is essential for understanding the hemispheric dichotomy of the Moon[1–3]. Cratering chronology established for the lunar nearside has been used to suggest long-lived volcanism on the farside of the Moon[3,4] but without sample verification. We describe two episodes of basaltic volcanism identified by Pb–Pb dating of basalt fragments returned by the Chang'e-6 mission. One high-Al basalt fragment, dated at $4,203 \pm 4$ million years ago (Ma), has a source $^{238}U/^{204}Pb$ ratio ($\mu$ value) of approximately 1,620, implying a KREEP-rich (K, rare earth elements and P) source for this oldest-known example of basaltic volcanism among returned samples. The main volcanic episode of the Chang'e-6 basalt documents a surprisingly young eruption age of $2,807 \pm 3$ Ma, which has not been observed on the nearside of the Moon. The initial Pb isotope compositions of these younger basalts indicate a derivation from a source with a $\mu$ value of approximately 360, indicating a KREEP-poor mantle source. Mare volcanism on the lunar farside thus persisted for over 1.4 billion years, even if the source was depleted in heat-producing elements. The consistency between the 2.8-billion-year basalt age and the crater-counting age indicates that the cratering chronology model established for the lunar nearside is also applicable to the farside of the Moon.

The enigmatic asymmetry between the nearside and the farside of the Moon—encompassing basalt distribution[5,6], topography[7], crustal thickness[8] and thorium (Th) concentration[9]—is a long-standing unresolved conundrum. Mare and cryptomare basalts cover approximately 18% of the lunar surface[2], with approximately 93% on the nearside and only approximately 7% on the farside[10]. A comprehensive understanding of the lunar hemispheric dichotomy requires knowledge of both the radiometric ages and petrogenesis of basaltic volcanism from both sides of the Moon. Studies of lunar basaltic samples returned by the Apollo, Luna and Chang'e-5 missions have established that nearside volcanism on the Moon occurred as early as 4.0 billion years ago (Ga)[11] and continued until at least 2 Ga (refs. 12,13). Small-scale volcanism may also have occurred on the nearside as late as approximately 120 million years ago (Ma), as recorded by volcanic glass beads from Chang'e-5 samples[14]. Whether such prolonged volcanic activity occurred on the lunar farside remains unclear. Despite the lower frequency and volume of volcanic eruptions on the farside[15], the chronology from impact crater size versus frequency distribution based on remote-sensing observations suggests that the age patterns of nearside and farside volcanism are potentially broadly similar[2–4]. However, the lack of samples returned from volcanic provinces on the lunar farside has heretofore precluded a robust comparison.

The South Pole–Aitken basin, located near the South Pole on the lunar farside[16], has the thinnest crust on the Moon. Most of the farside basalts are concentrated in this region[2,10]. China's Chang'e-6 mission, the first lunar mission to return samples from the farside of the Moon, landed on 2 June 2024 on the southern mare of the Apollo basin in the north-eastern South Pole–Aitken basin[17] (Extended Data Fig. 1). Chang'e-6 successfully returned 1,935.3 g of lunar soil[17], which provides a unique opportunity to study farside volcanism. The Chang'e-6 samples analysed in this study include two aliquots of soils scooped from the surface (samples CE6C0100YJFM002 of 2 g and CE6C0200YJFM001 of 3 g) allocated by the China National Space Administration.

Approximately 400 lithic fragments (over 300 µm in size) were randomly hand-picked and embedded in epoxy mounts. Micropetrographic observations revealed that basalt fragments account for approximately 30% of the selected lithic fragments. The basalt fragments exhibit a range of crystal sizes, from under 10 µm to 500 µm. Most of the basalt fragments (91%) show medium- to coarse-grained subophitic and poikilitic textures, with minor occurrences (9%) of porphyritic texture composed of phenocrysts larger than 300 µm and a cryptocrystalline to fine-grained matrix (Supplementary Fig. 1). Despite the variation in texture, the major mineral constituents of the basalt fragments are similar, comprising clinopyroxene, plagioclase and ilmenite, with minor troilite (Fig. 1 and Extended Data Fig. 2). Most ilmenite crystals cross-cut other major minerals. This crystallization sequence is common in low-titanium (Ti) basalts. Euhedral to subhedral phosphate minerals (such as apatite, merrillite and changesite-[Y])

[1]State Key Laboratory of Lithospheric and Environmental Coevolution, Institute of Geology and Geophysics, Chinese Academy of Sciences, Beijing, China. [2]College of Earth and Planetary Sciences, University of Chinese Academy of Sciences, Beijing, China. [3]Key Laboratory of Lunar and Deep Space Exploration, National Astronomical Observatories, Chinese Academy of Sciences, Beijing, China. ✉e-mail: liqiuli@mail.iggcas.ac.cn

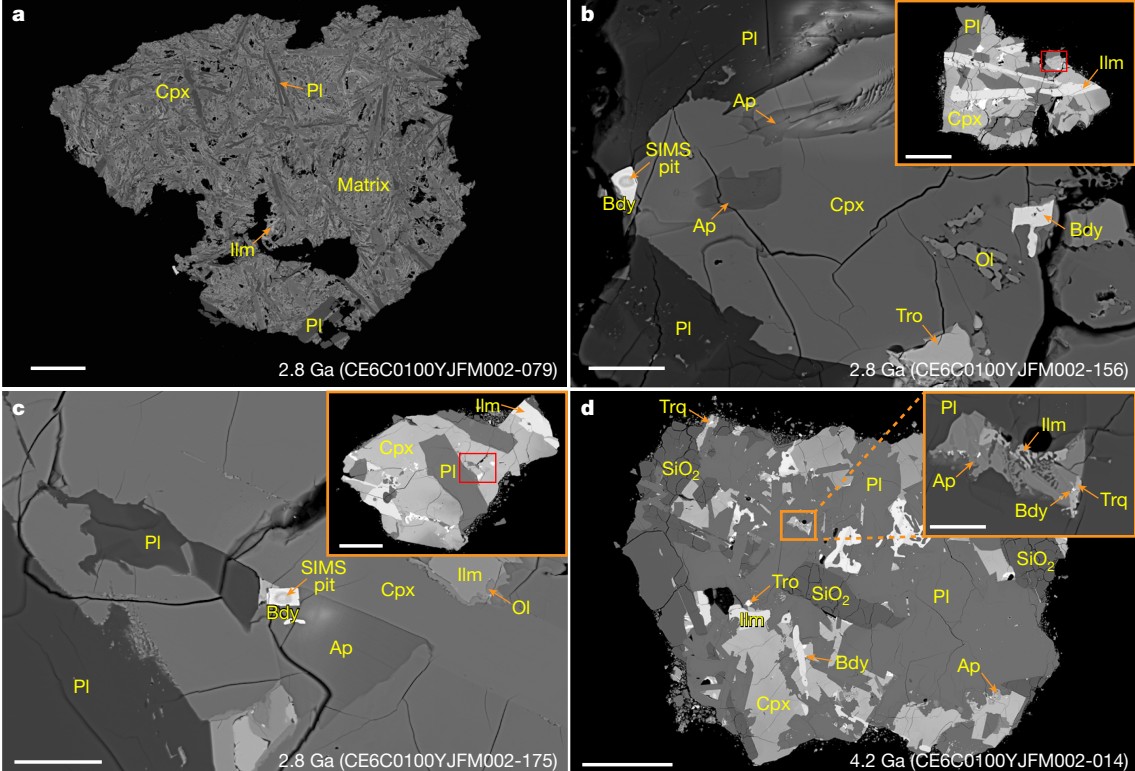

**Fig. 1 | BSE images showing micropetrographic features of representative dated basalt fragments from Chang'e-6 samples. a**, Porphyritic basaltic fragment consisting of large phenocrysts and a cryptocrystalline to fine-grained matrix. **b**, Subhedral baddeleyite and apatite occur as intergranular phases and inclusions accompanied by Fe-rich olivine, troilite, clinopyroxene and plagioclase in a subophitic clast. **c**, Euhedral baddeleyite occurs as an intergranular phase associated with apatite and plagioclase in a poikilitic fragment. **d**, Micropetrographic features of a high-Al basalt fragment. The inset shows an intergrowth texture of baddeleyite, tranquillityite, apatite, ilmenite and clinopyroxene. Pits on the baddeleyite crystals indicate the in situ analytical spots for SIMS analysis. The areas of **b** and **c** where minerals were dated in the 2.8 Ga basalt fragments are highlighted with red rectangles in the corresponding insets. The Zr-bearing minerals (baddeleyite and tranquillityite) used for dating the 4.2 Ga basalt fragment are shown in the inset of **d**. The formation age, sample name and grain number are indicated in the bottom right corner of each image. Yellow arrows are used to indicate the location of fine-grained minerals. Four other representative basalt fragments are presented in Extended Data Fig. 2. BSE images of all 108 dated basalt fragments can be found in Supplementary Fig. 1. Ap, apatite; Bdy, baddeleyite; Cpx, clinopyroxene; Ilm, ilmenite; Ol, olivine; Pl, plagioclase; Tro, troilite; Trq, tranquillityite. Scale bars, 150 μm (**a**, inset of **b**, inset of **c**, **d**), 20 μm (**b**, **c**), 15 μm (inset of **d**).

and anhedral cristobalite are commonly found in basalt fragments with subophitic and poikilitic textures but are rare in porphyritic-type basalt fragments (Fig. 1 and Extended Data Fig. 2). Minor zirconium (Zr)-bearing minerals (such as baddeleyite, zirconolite and tranquillityite) occur as fine-grained euhedral to subhedral crystals, mostly less than 3 μm in size, and are present only in subophitic and poikilitic basalt fragments being absent in porphyritic fragments. These Zr-bearing minerals are often associated or intergrown with apatite, troilite, iron (Fe)-rich olivine and mesostasis, suggesting that they formed during the final crystallization stage of the basaltic magma (Fig. 1 and Extended Data Fig. 2).

## Ages of Chang'e-6 basalt fragments

Radioisotopic dating was conducted on 108 basalt fragments, including nine porphyritic, 45 subophitic and 54 poikilitic fragments (Extended Data Table 1). All the basalt fragments in this study exhibit typical magmatic textures, allowing their ages to be interpreted as the formation ages associated with basaltic volcanism. The Pb isotopic compositions of various mineral phases in the Chang'e-6 basalt fragments were determined using a CAMECA IMS 1280HR secondary-ion mass spectrometer (SIMS; Supplementary Table 1). Given that the minerals have various grain sizes and Pb concentrations, primary oxygen beams with spot sizes of roughly 3 and 30 μm in diameter were used for Zr-bearing minerals and other phases, respectively (Methods).

Pb isotopic compositions were measured to construct the Pb–Pb leftmost isochron[12,18], the *y* intercepts of which represent the radiogenic $^{207}Pb/^{206}Pb$ ratios of basalt fragments, which can be converted to Pb–Pb ages.

Initially, we treated each basalt fragment as an individual sample (Extended Data Figs. 3 and 4), with potentially different ages and mantle sources. For the poikilitic fragments, Pb isotope analyses of three fragments yielded isochrons with consistent slopes around 162 and ages of 2,811 ± 44 Ma (Extended Data Fig. 3a,b), 2,811 ± 7 Ma (Extended Data Fig. 3c,d) and 2,762 ± 36 Ma (Extended Data Fig. 3e,f). Pb isotope compositions obtained from the other 50 poikilitic fragments also aligned with the approximately 2.8 Ga isochron, suggesting that they may be from the same episode of parent magma. Taken together, the analyses of various mineral phases with negligible terrestrial Pb contamination from 53 poikilitic fragments allowed us to construct a combined leftmost isochron yielding a Pb–Pb age of 2,807 ± 3 Ma (95% confidence level here and hereafter, except where otherwise noted; Extended Data Fig. 5a,b). We applied a similar procedure to the subophitic fragments. A leftmost isochron constructed from 18 analyses on a large subophitic fragment yielded an age of 2,813 ± 23 Ma (Extended Data Fig. 3g,h) and slope of 162 ± 9, closely matching that of the poikilitic fragments. Together with the Pb isotope compositions of the other 44 subophitic fragments, a leftmost isochron yielded a Pb–Pb age of 2,805 ± 4 Ma (Extended Data Fig. 5c,d). For porphyritic basalt fragments lacking visible Zr-bearing minerals, Pb isotope analyses were performed on

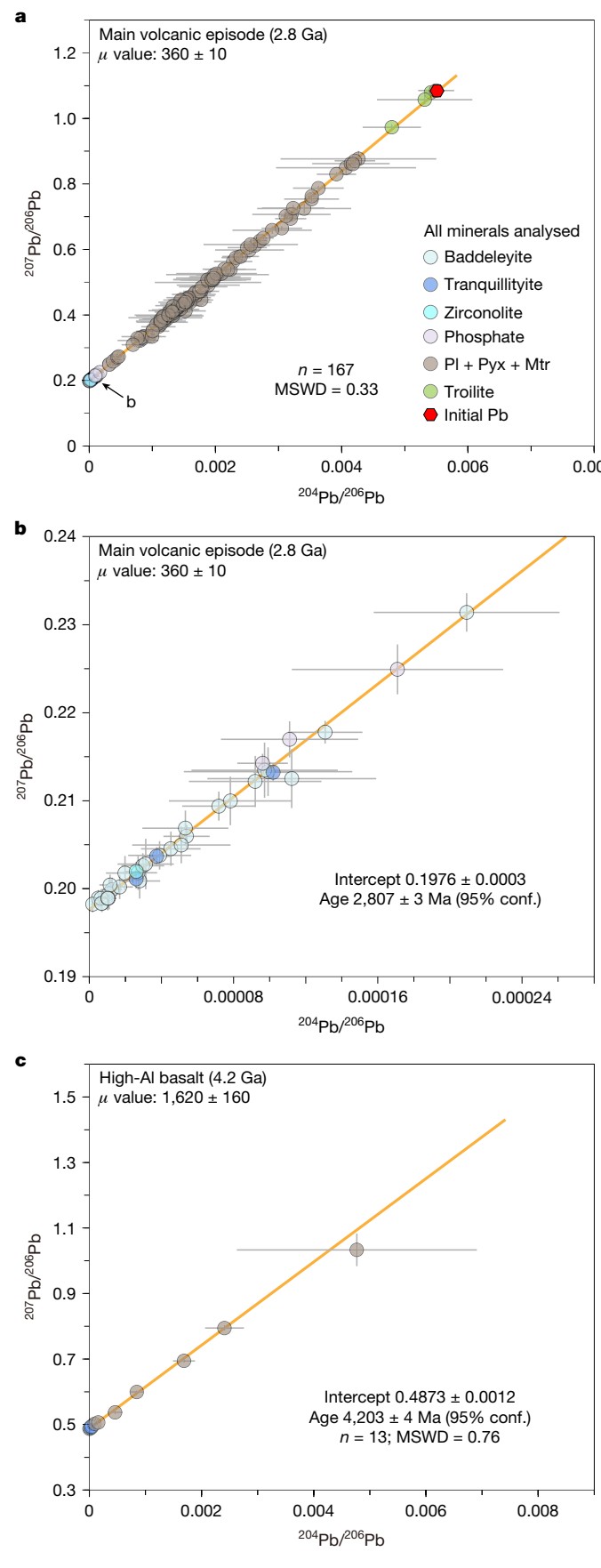

**Fig. 2 | Pb–Pb isochrons for the two episodes of basaltic volcanism identified in the Chang'e-6 basalts. a**, Integrated Pb–Pb isochron of the 2.8 Ga basalt showing the mixing between the radiogenic $^{207}Pb/^{206}Pb$ represented by the *y* intercept and the initial Pb composition ($^{204}Pb/^{206}Pb = 0.00550 ± 0.00014$ and $^{207}Pb/^{206}Pb = 1.085 ± 0.003$). **b**, Enlarged section near the *y* intercept of the isochron in **a**, highlighting measurements of Zr-bearing minerals and apatite. **c**, Integrated Pb–Pb isochron of the 4.2 Ga basalt. The *μ* values ($^{238}U/^{204}Pb$ ratios) of the mantle sources of two episodes of basalt volcanism are presented in the top left corners. Error bars represent 2 standard errors. The yellow lines are the best-fitting isochrons integrated from analyses with negligible terrestrial Pb contamination. The uncertainties of isochron ages are quoted at the 95% confidence level. conf., confidence level; MSWD, mean square weighted deviation; Pyx, pyroxene; Mtr, matrix.

Data Fig. 4). Overall, Pb isotope analyses (*n* = 79) of nine porphyritic clasts yielded a leftmost isochron Pb–Pb age of 2,815 ± 72 Ma (Extended Data Fig. 5e,f) with a slope like that from dating the poikilitic and sub-ophitic basalt fragments within uncertainties, indicating that they were formed during the same volcanic episode despite the different petrographic textures. Taken together, the 167 analyses of various mineral phases with negligible terrestrial Pb contamination allowed us to construct an integrated leftmost isochron yielding a Pb–Pb age of 2,807 ± 3 Ma (Fig. 2a,b). Most (approximately 99%) of the basalt fragments studied have a consistent formation age of approximately 2.8 Ga, which was taken as representing the age of the main volcanic episode at the Chang'e-6 landing site. This age is an intermediate value between that of the approximately 2.0 Ga Chang'e-5 basalts and the 3.8–3.2 Ga Apollo basalts.

The basalt fragment CE6C0100YJFM002-014 displays distinct petrographic characteristics with high proportions of plagioclase (Fig. 1d). Plagioclase crystals in this fragment show an initial increase and then a decrease of MgO with decreasing anorthite content (Extended Data Fig. 6a and Supplementary Table 2). This feature implies clinopyroxene crystallization following plagioclase, a typical sequence in high-aluminium (Al) basalt[19], which is also supported by the decreasing $Al_2O_3$ content with decreasing Mg# (= molar Mg/(Mg + Fe)) of clinopyroxene (Extended Data Fig. 6b). The leftmost isochron constructed for baddeleyite, tranquillityite and silicate minerals yielded a *y* intercept for the radiogenic $^{207}Pb/^{206}Pb$ ratio of 0.48729 ± 0.0012 corresponding to a Pb–Pb age of 4,203 ± 4 Ma (Fig. 2c). To date, this is the oldest returned lunar high-Al basalt sample with a precise age determination, which has an approximately 0.1% uncertainty. Its age is comparable to the nearside 4.3–4.0 Ga high-Al basalt returned by Apollo missions[20] but younger than the approximately 4.36 Ga high-Al volcanism documented in lunar meteorite Kalahari 009 (a monomict basaltic breccia). Spectral data reveal wide distributions of high-Al basalt on both sides of the Moon[21,22], implying a potential nearside ejection origin. However, given that the high-Al basalt fragment studied exhibits a pristine magmatic texture with no evidence of impact-induced modification, we regard it as a local product of the lunar farside rather than ejecta from the nearside. Notably, the crater-counting age of the cryptomare region on the south of the Chang'e-6 landing site[23,24] is approximately 4.05 Ga (ref. 4), consistent with the age of the high-Al basalt, within the uncertainty. Combined with remote-sensing observations showing that volcanic materials beneath the cryptomare are plagioclase-rich, the high-Al basalt fragment documented here most probably originated from this nearby cryptomare region. Therefore, the formation ages of 2.8 Ga for the main Chang'e-6 volcanic episode and 4.2 Ga for the high-Al basalt imply that volcanism on the lunar farside spanned at least 1.4 billion years.

## Diverse mantle sources

Two main factors are thought to control the lunar dichotomy in terms of mare volcanism: (1) crust thickness[25] and (2) the distribution of radioactive heat-producing elements[26]. Tracing the geochemical

pyroxene, plagioclase and the fine-grained matrix. The results for any single fragment defined an imprecise isochron, nonetheless they fell into the aforementioned approximately 2.8 Ga isochron (Extended

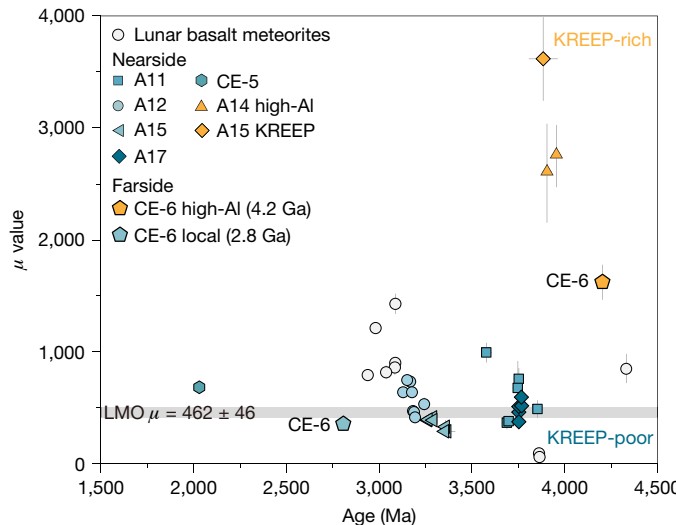

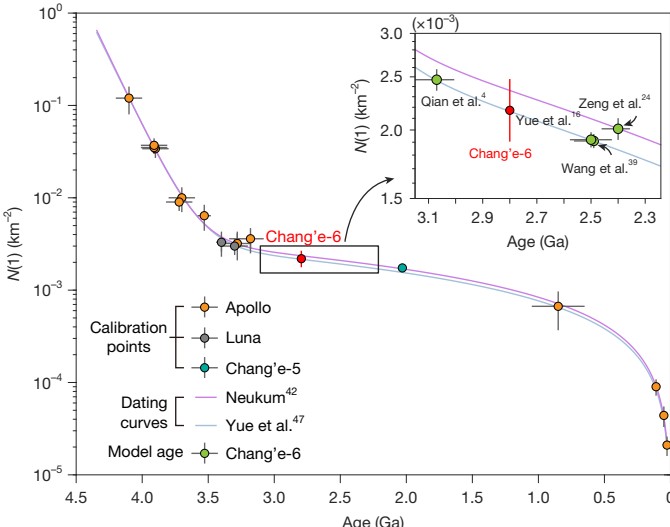

**Fig. 3 | Mantle source $\mu$ values over time for returned nearside and farside lunar basalts and lunar basalt meteorites.** The $\mu$ value of the lunar magma ocean (LMO) representing the bulk Moon is indicated by the light grey band. Data are sourced from refs. 11,12,18,34–36,48,49. Error bars represent 2 standard errors. A11, Apollo 11 high-Ti basalts; A12, Apollo 12 low-Ti basalts; A14 high-Al, Apollo 14 high-Al basalts; A15, Apollo low-Ti basalts; A15 KREEP, Apollo 15 KREEP basalts; A17, Apollo 17 high-Ti basalts; CE-5, Chang'e-5 low-Ti basalts; CE-6 high-Al, Chang'e-6 high-Al basalts of approximately 4.2 Ga; CE-6 local, Chang'e-6 local low-Ti basalts of approximately 2.8 Ga.

**Fig. 4 | Lunar crater-counting chronology compared to the critical reference point of the radioisotope age of the Chang'e-6 local basalt.** The yellow, grey and cyan circles represent calibration points respectively derived from Apollo, Luna and Chang'e-5 samples[39,44,50]. The purple and light blue curves are the previous crater-counting chronology function of ref. 39 and the updated function incorporating the radioisotope age of the Chang'e-5 basalt[12,44], respectively. The red circle represents the radioisotope age of the main volcanic phase of the returned Chang'e-6 samples. The red vertical line shows the range of published $N(1)$ values of the cumulative frequencies of craters over 1 km in diameter in the Chang'e-6 landing region indicated by green points[4,45–47]. The error bars of $N(1)$ values represent $1\sigma$ uncertainty, whereas the error bars of age represent 95% confidence. Inset, Zoom-in of the boxed area.

characteristics of the mantle sources of the two Chang'e-6 farside mare volcanic episodes informs not only the internal evolution of the lunar farside but also the potential of a dichotomy within the lunar mantle. The initial Pb isotopic compositions of the basalts and corresponding time-independent $\mu$ values ($^{238}U/^{204}Pb$ ratios) of their mantle sources can be used to evaluate the enrichment or depletion of KREEP components (high concentrations of potassium (K), rare earth elements (REE) and phosphorous (P)) in the mantle sources.

Using a two-stage model for the lunar Pb isotopic evolution[18] (Methods), the calculated $\mu$ values of the mantle sources for the two volcanic episodes identified are notably different, indicating two diverse mantle sources with varying degrees of KREEP component hybridization. The mantle source of the 4.2 Ga high-Al basalt exhibits a $\mu$ value of 1,620 ± 160, lower than that of the approximately 3.95–3.90 Ga nearside high-Al basalts or approximately 3.88 Ga KREEP basalt[11] but significantly higher than those of mare basalts (Fig. 3). The compositions of this basalt fragment and its high-$\mu$ source features suggest a plagioclase-involved, KREEP-rich mantle source. Several mechanisms may account for this hybrid mantle source, such as mantle overturn or a giant impact[27,28], which could transport KREEP material into the mantle or cause the assimilation of KREEP and plagioclase-rich materials as magma ascends to the lunar surface[29]. In any case, the formation of the 4.2 Ga high-Al basalt implies that portions of the KREEP component may have persisted in the farside mantle despite experiencing the older than the 4.2 Ga South Pole-Aitken giant impact[30–33].

The apparent progressive increase in $\mu$ values for mantle sources from 300–1,400 with decreasing formation age from 3.4–3.0 Ga, as recorded in Apollo low-Ti basalts and low-Ti and very-low-Ti basaltic meteorites[11,18,34], implies a progressive contribution of KREEP-like components in the mantle sources of younger basalts[34]. This trend does not extend to samples as recent as 2.0 Ga, as shown by the Chang'e-5 basalt with a low-$\mu$ source feature[12]. We calculated the $\mu$ value of the mantle source of the 2.8 Ga Chang'e-6 basalt to be 360 ± 10, one of the lowest $\mu$ values reported for lunar maria (Fig. 3). Thus, this provides a constraint on the evolving composition of lunar mantle sources between 3 and 2 Ga and suggests that the main volcanic episode at the Chang'e-6

landing site was produced by the melting of a KREEP-poor source. Combined with the mantle source features of all returned basalts, the high-$\mu$ mantle sources dominated the formation of early basaltic volcanism (older than 3.9 Ga)[11,18], whereas subsequent mare volcanism was predominantly produced by the melting of intermediate-$\mu$ and low-$\mu$ mantle sources[11,12,34–37] (Fig. 3). This stark contrast indicates that the concentration of radioactive heat-producing elements was probably not a decisive factor in the mantle melting responsible for young lunar volcanism. The wide range of observed $\mu$ values of farside mantle sources identified in this study encompasses nearly most of those of nearside mantle sources[11,12,18,34], except those of three special KREEP and high-Al basalts, and implies that there are comparable hemispheric mantle compositions on either side of the Moon. Instead then, the crustal thickness may, therefore, be the key factor controlling the volcanic asymmetry between the nearside and farside of the Moon.

## Anchor point for cratering chronology

For the unexplored regions of the Moon, age estimates are mostly obtained using lunar crater-counting chronology[5,38,39], which has even been extended to other terrestrial bodies within the inner Solar System[40–43]. Radiometric ages acquired for returned lunar samples, which anchor the relative dating of the crater-counting chronology model in absolute time, constitute the cornerstone of the method[44]. However, there is a large temporal gap in calibration anchor points for the cratering curve between approximately 3.2 and 2.0 Ga, with the 2.8 Ga Chang'e-6 basalts filling in this critical gap. Furthermore, the cratering model has not yet been validated with samples from the lunar farside, particularly regarding the debated issue of whether the impact probabilities on the farside and nearside are comparable or not. Therefore, the radiometric age of the main volcanic episode of the returned Chang'e-6 samples can be used to evaluate the applicability

of the lunar crater-counting method on both sides of the Moon and, further, to compare the impact flux between the nearside and farside.

Remote-sensing observations suggest that basalts in the southern Apollo basin can be classified into low-Ti and medium-Ti types[4,45]. The Chang'e-6 landing site is in the medium-Ti region, and the suggested crater-counting ages are approximately 2.40 Ga (ref. 45) using the model of Neukum[39] and approximately 2.49 Ga (ref. 46), approximately 2.50 Ga (ref. 47) and approximately 3.07 Ga (ref. 4) using the recently updated model of Yue et al.[44] with the calibration of the Chang'e-5 anchor point (Fig. 4). The eruption age of the local basaltic volcanism at the Chang'e-6 landing site is dated at approximately 2.8 Ga, which aligns with the median range of published crater-counting model ages, within 10% uncertainties, using the model of Yue et al.[44]. This consistency suggests that the cratering chronology function established for the nearside of the Moon is also applicable to the lunar farside. Thus, the petrological analysis and radioisotope ages of the Chang'e-6 samples established in this study could be used to refine the current lunar crater-counting chronology function by providing another critical calibration point, thereby improving its precision.

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

# Methods

All sample preparation tasks and analyses, including scanning electron microscopy (SEM), electron microprobe analysis (EMPA) and SIMS, were conducted at the Institute of Geology and Geophysics, Chinese Academy of Sciences, in Beijing, China.

## SEM analysis
The studied Chang'e-6 basalt fragments were embedded in epoxy mounts and then polished using a grinder with fine diamond pastes (grit sizes of 1 and 0.25 μm). For the SEM analysis, the samples were coated with an approximately 8 nm carbon layer. High-resolution backscattered electron (BSE) imaging and semi-quantitative energy-dispersive spectroscopy analyses were performed using a Thermo Scientific Apreo SEM and a Zeiss Gemini 450 field-emission environmental SEM equipped with an energy-dispersive spectroscopic detector. The measurements were performed using a 15 kV accelerating voltage and a 2.0 nA current, with a working distance of approximately 8.5 mm. The phosphate (apatite) and Zr-bearing minerals (baddeleyite, tranquillityite and zirconolite) were identified by energy-dispersive spectroscopy.

## EMPA
The compositions of the major elements in pyroxene and plagioclase in the high-Al basalt fragment were measured using a JEOL JXA 8100 electron microprobe analyser. The analytical conditions were as follows: 15 kV accelerating voltage, 20 nA beam current, 3 μm spot diameter and 20 s peak counting time with 10 s counting time at the lower and upper background positions. The elemental data were calibrated using a series of natural minerals and synthetic materials (Supplementary Table 3). All test data were corrected online using a modified ZAF correction procedure. The detection limits ($3\sigma$) ranged from 0.01 to 0.03 wt%. The precisions for major (more than 1.0 wt%) and minor (0.1–1.0 wt%) elements were better than 1.5% and 5.0%, respectively.

## SIMS analysis
The overall analysis conditions followed the procedure of Li et al.[12] in their study of Chang'e-5 basalts. The target selection strategy focused on identifying phases that could contain both initial lead (Pb) or radiogenic Pb produced from the in situ decay of uranium (U) after the sample had crystallized (mostly Zr-bearing phases). These two phases yielded distinct Pb isotopic ratios, which are essential for constructing the isochron and determining a precise age. The Pb isotopic compositions (Supplementary Table 1) were determined over two analytical sessions using a CAMECA IMS 1280HR ion microprobe equipped with an updated radio-frequency oxygen source. To minimize contamination, the mounts containing basalt fragments were cleaned with a fine diamond paste (0.25 μm) and ethanol before being coated with approximately 20 nm carbon.

The first analytical session focused on measuring Pb isotopes in Zr-bearing minerals. A Gaussian illumination mode was employed to focus a primary $^{16}O^-$ beam to a size of approximately 3 μm (ref. 51; Fig. 1 and Extended Data Fig. 2), with an accelerating potential of −13 kV. The beam size remained stable over extended use, with intensities of approximately 120 pA. Multi-collector mode with four electronic multipliers was used to measure $^{204}Pb^+$ (L2), $^{206}Pb^+$ (C), $^{207}Pb^+$ (H1) and $^{96}Zr_2^{16}O_2^+$ (H2). Exit slit 3 was used, which provided a mass resolving power of 8,000 (50% peak height). Before each analysis, a $^{16}O^-$ primary beam with a 10 nA intensity was used for pre-sputtering over 120 s. Ion images of $^{96}Zr_2^{16}O_2^+$ and Pb isotopes within a 25 μm × 25 μm area were generated to precisely locate the target minerals. The $^{206}Pb$ signal was used for peak centring reference. Each measurement consisted of 100 cycles each lasting 8 s, with a total analytical duration of approximately 17 min. High-purity oxygen gas was introduced onto the sample surface to enhance the Pb$^+$ yield to over 15 counts per second (cps) ppm$^{-1}$ nA$^{-1}$ (ref. 52) using an O$^-$ primary beam according to the M257 zircon standard (561 Ma,

840 ppm U)[53]. National Institute of Standards and Technology (NIST) standard reference material 610 (SRM610) glass[54] was used to calibrate the relative yield of different electronic multipliers and to evaluate the external reproducibility. Based on 30 analyses of NIST SRM610 glass (Supplementary Table 4) under the same analytical conditions, the $^{207}Pb/^{206}Pb$ measurements had a relative standard deviation of 0.57%, with $^{207}Pb$ intensities averaging 82 cps. The potential SIMS instrumental mass fractionation of Pb isotopes, approximately 0.2% (ref. 55), was propagated to the uncertainty of the single-spot $^{207}Pb/^{206}Pb$ analysis.

The second session focused on determining the main isochron and initial Pb composition of the essential minerals, including plagioclase, pyroxene and troilite, and the matrix. A Köhler illumination mode was used to generate a primary $^{16}O_2^-$ beam of approximately 32 nA, focused to an approximately 30 μm spot size. Before each measurement, a 25 μm area around the target spot was raster-scanned for 120 s to remove the coating and minimize potential contamination. A dynamic multi-collector mode with four electronic multipliers was used to measure $^{28}Si^{30}Si^{16}O^+$ in the first step as reference peaks for essential minerals, $^{204}Pb^+$ (L2), $^{206}Pb^+$ (C), $^{207}Pb^+$ (H1) and $^{208}Pb^+$ (H2) with 30 s counting time in the second step, and $^{238}UO^+$ (H2) in the third step with 2 s counting time. Each measurement consisted of 20–30 cycles. Based on 35 analyses of NIST SRM 610 glass (Supplementary Table 4) under identical conditions, the $^{207}Pb/^{206}Pb$ measurements had a relative standard deviation of 0.2% with $^{207}Pb$ intensity averaging 15,000 cps. Background counts for each channel (Supplementary Table 5) were recorded at regular intervals during each session using deflector and aperture settings that effectively blanked both the primary and residual secondary beams. The numbers of SIMS Pb isotope analyses for each type of basalt fragments used in this study are summarized in Extended Data Table 1.

## Data processing
The data were processed using in-house SIMS data reduction spreadsheets and the Excel add-in Isoplot (ref. 56). The leftmost Pb–Pb isochrons were constructed following the same method applied in previous SIMS studies[12,18]. The leftmost boundary of the three-component mixing triangle defined by the initial Pb, radiogenic Pb and terrestrial Pb (ref. 57) compositions forms an isochron, which was determined by iteratively filtering the data to achieve the steepest statistically significant weighted regression. The probable sources of the terrestrial Pb contamination are the coating materials (with carbon being preferable over gold for reducing terrestrial Pb contamination) and residual polishing materials, which could sink into grain boundaries and cracks. The $^{207}Pb/^{206}Pb$ ratio of the initial Pb was estimated from spots with the highest $^{207}Pb/^{206}Pb$ and near-zero $^{238}UO^+/^{208}Pb^+$ (Extended Data Fig. 7). For the age calculations, we used the decay constants of $1.55125 \times 10^{-9}$ for $^{238}U$ and $9.8485 \times 10^{-11}$ for $^{235}U$ and the $^{238}U/^{235}U$ ratio of 137.818 (ref. 58).

The $\mu$ values ($^{238}U/^{204}Pb$) of mantle sources were determined using the lunar Pb isotope evolution model, following the procedures outlined by Snape et al.[18] and Li et al.[12]. The crystallization of the lunar magma ocean led to the separation of an isotopically homogeneous lunar mantle into distinct silicate reservoirs, which subsequently became the sources of chemically diverse basaltic rocks. The source $\mu$ value and initial Pb isotopic compositions can be predicted by calculating the intercept between the palaeo-isochrons derived from the model differentiation point and the primary sample isochrons. Alternatively, the $\mu$ values can be determined using the initial $^{204}Pb/^{206}Pb$ ratios of the basalts. The parameters used include the model starting Pb isotope composition of $^{204}Pb/^{206}Pb = 9.307$ based on Canyon Diablo troilite[59], the model starting time of 4,567 Ma for the Solar System and 4,500 Ma for the Moon's formation, the $\mu$ value for bulk lunar silicate of $462 \pm 46$ (ref. 18) and the time $4,376 \pm 18$ Ma for the differentiation that marks the formation of distinct silicate reservoirs[18]. We applied both approaches to the 2.8 Ga local basalts and obtained consistent $\mu$ values, demonstrating the reliability of the results. For the 4.2 Ga basalt, only the first approach was used because no precise initial Pb composition was determined.

## Data availability

All data generated in this study are included in Supplementary Tables 1–5 and are available on Zenodo at https://doi.org/10.5281/zenodo.14053388 (ref. 60). Source data are provided with this paper.

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

**Acknowledgements** We thank all the staff of China's Chang'e Lunar Exploration Project for their contributions in returning the lunar samples. The samples in this study were allocated by the China National Space Administration. We thank R. Mitchell for constructive comments on editing the manuscript. We appreciate Y. Chen, W. Yang, H.-J. Hui, S. Gou, S. Hu, Y.-T. Lin, C.-L. Li and F.-Y. Wu for helpful discussions. We thank S.-H. Zhao for logistical support. We thank X.-X. Ling, J. Li, G.-Q. Tang, D. Zhang, L.-H. Jia, J.-Y. Yuan and L.-X. Gu for their help with the SIMS, EMPA and SEM analyses. This study was funded by the National Natural Science Foundation of China (Grant Nos. 42225301 and 42241105), the Strategy Priority Research Program (Category B) of the Chinese Academy of Sciences (Grant No. XDB0710000) and the Key Research Program of the Institute of Geology and Geophysics, Chinese Academy of Sciences (Grant No. IGGCAS-202401).

**Author contributions** Q.-L.L. and X.-H.L. conceived and supervised this project. Q.W.L.Z., Q.-L.L. and M.-H.Y. wrote the manuscript with input from X.-H.L. and Z.-Y.Y. H.-X.M. conducted the grain separation and sample mounting. Q.W.L.Z., M.-H.Y., Q.Z., S.-H.Y. and X.T. performed the SEM analyses. G.-L.Z. and X.R. performed the EMPA analyses. M.-H.Y., Q.-L.L., Y.L., Q.W.L.Z. and L.-Y.C. conducted the SIMS analyses. Q.-L.L., Q.W.L.Z., M.-H.Y., L.-Y.C. and X.-H.L. performed the SIMS data processing. All authors were involved in interpreting the data.

**Competing interests** The authors declare no competing interests.

**Additional information**
**Correspondence and requests for materials** should be addressed to Qiu-Li Li.

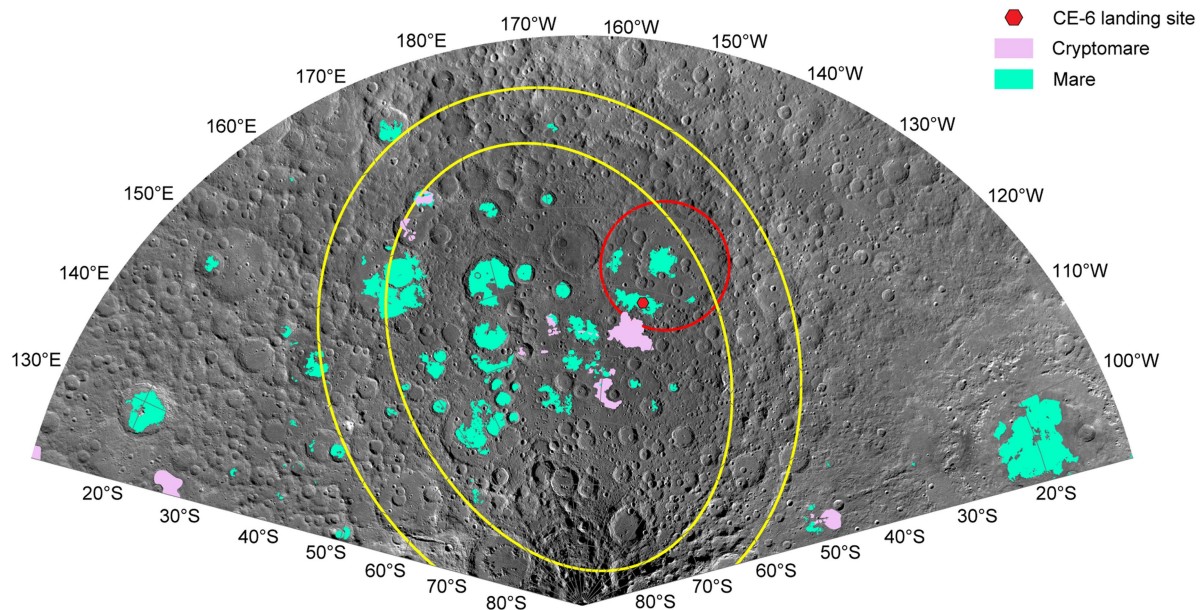

**Extended Data Fig. 1 | Mare and cryptomare distribution within and surrounding the SPA basin.** The cryptomare boundaries are sourced from ref. 23 and the mare boundaries are from ref. 6. The two yellow ellipses are the inner and outer rings of the SPA basin[16], while the red circle is the Apollo crater rim.

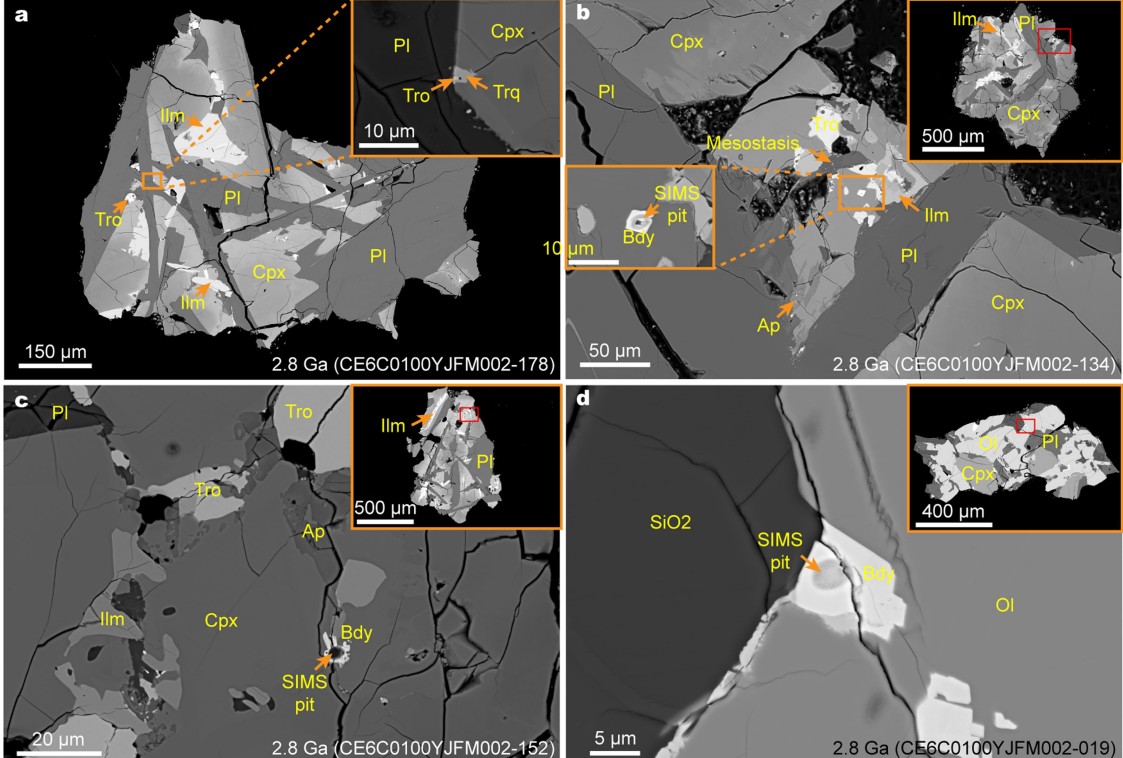

**Extended Data Fig. 2 | Back-scattered electron images showing the micropetrographic features of representative dated basalt fragments from Chang'e-6 samples. a**, A subophitic basalt fragment containing a fine-grained tranquillityite crystal, with the orange rectangle expanded in the inset. **b**, A square-shaped baddeleyite crystal occurs as inclusion in Fe-rich clinopyroxene, surrounded by tiny ilmenite, troilite, and mesostasis in a poikilitic fragment. **c**, Subhedral baddeleyite and apatite occur as intergranular phases between clinopyroxene, troilite, and ilmenite in a subophitic basalt. **d**, Euhedral baddeleyite occurs as an intergranular phase associated with $SiO_2$ and Fe-rich olivine in a poikilitic fragment. Pits on baddeleyite crystals indicate the *in situ* analytical spots for SIMS analyses. The tranquillityite crystal used for dating in **a** is shown in the inset. The areas of dated minerals in **b**–**d** are outlined with red rectangles in the corresponding insets. Ap, apatite; Bdy, baddeleyite; Trq, tranquillityite; Cpx, clinopyroxene; Pl, plagioclase; Ilm, ilmenite; Ol, olivine; Tro, troilite. The sample names and grain numbers are displayed in the bottom right corner of each panel.

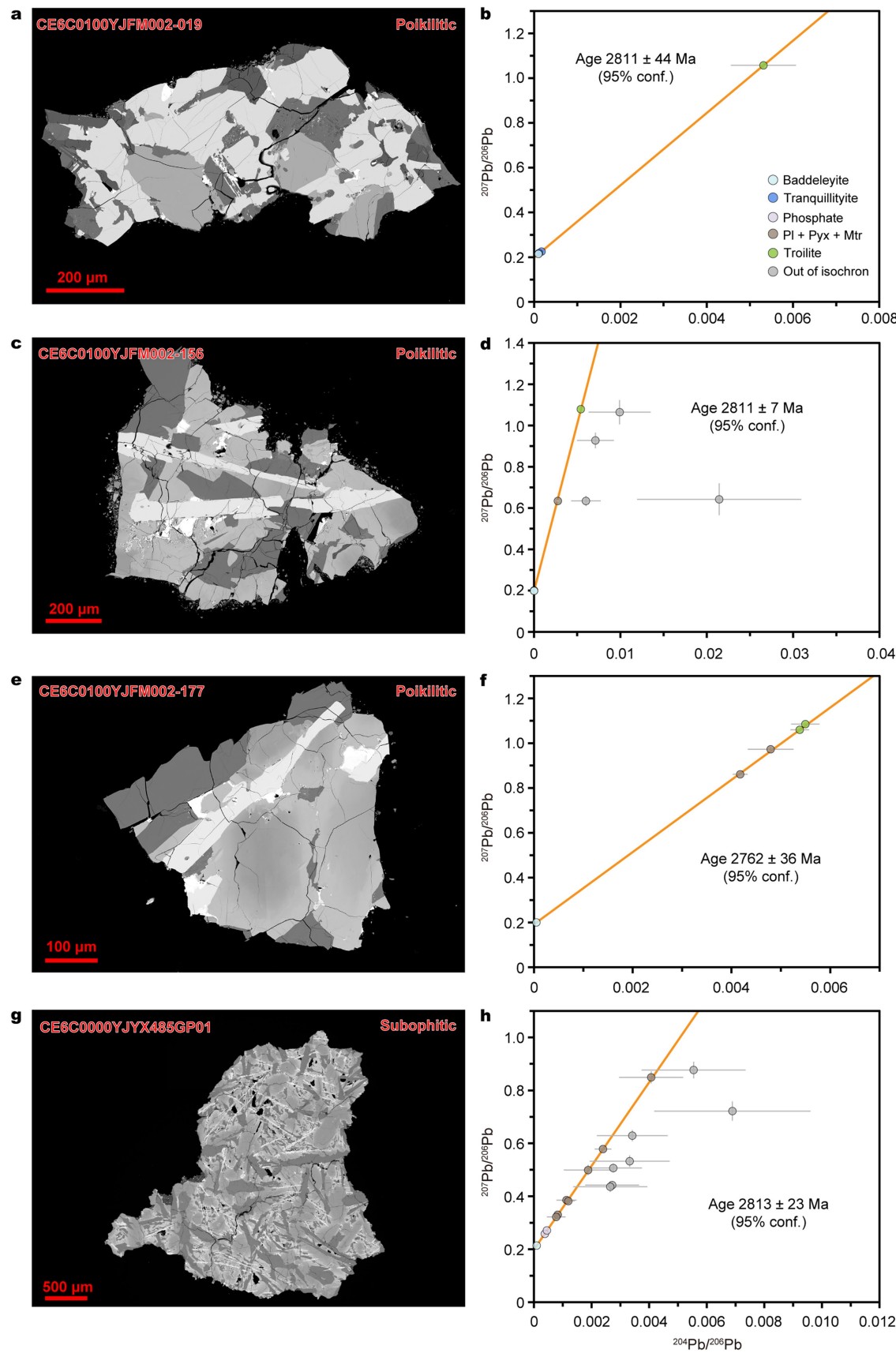

**Extended Data Fig. 3 | Pb-Pb isochrons for representative poikilitic and subophitic basalt fragments in Chang'e-6 samples.** The left four panels (a, c, e, and g) show BSE images of three poikilitic and one subophitic basalt fragment. The right four panels (b, d, f, and h) present corresponding Pb-Pb isochrons with consistent ages of ca. 2.8 Ga. Spots that deviate from the isochron are shown as grey circles and are excluded from the regression of the leftmost isochrons. Error bars represent 2 s.e. (standard error). The uncertainties of isochron ages are reported at the 95% confidence (conf.) level. Pl, plagioclase; Pyx, pyroxene; Mtr, matrix.

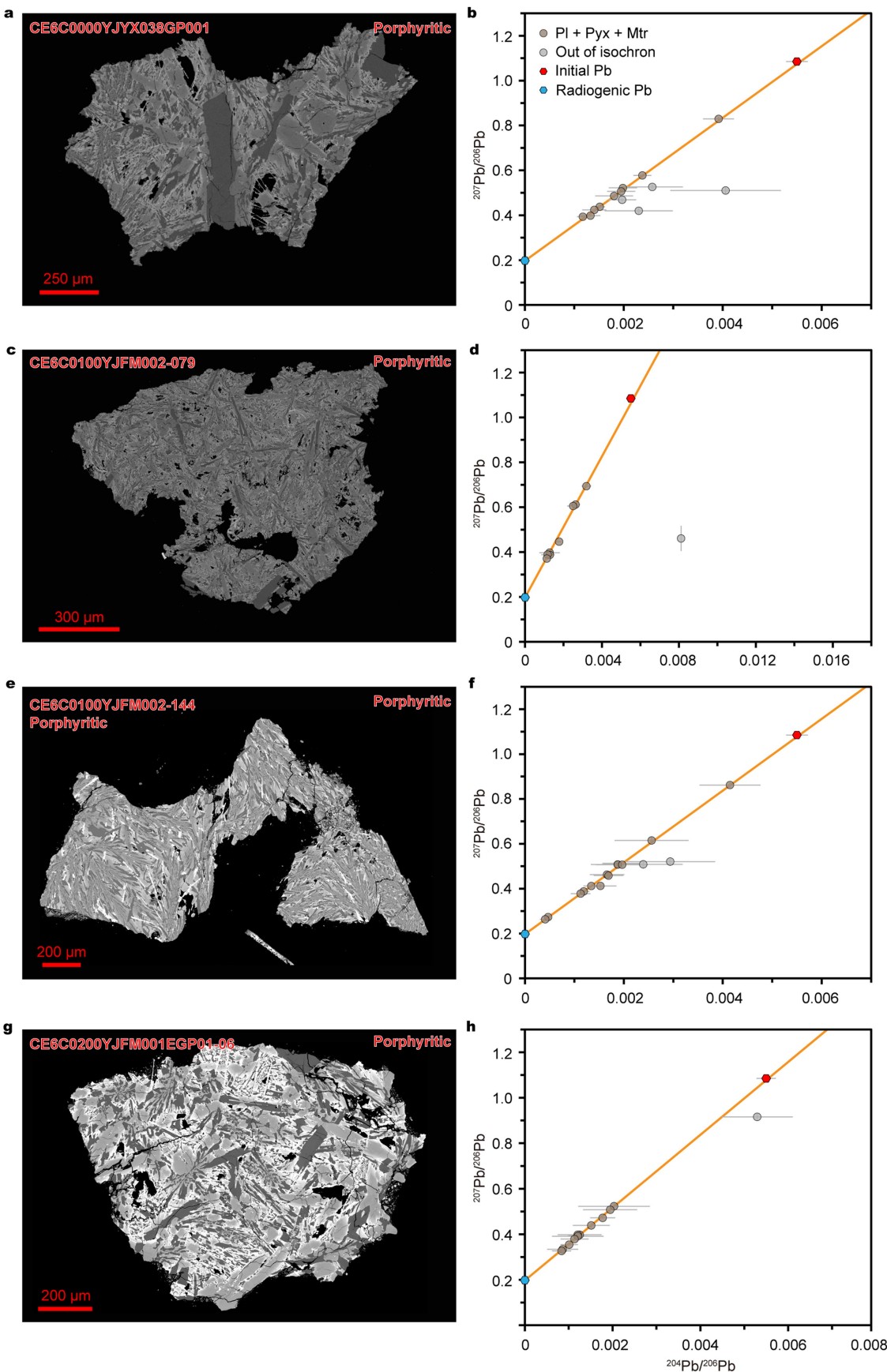

**Extended Data Fig. 4** | See next page for caption.

**Extended Data Fig. 4 | Pb-Pb isochrons for representative porphyritic basalt fragments in Chang'e-6 samples.** The left four panels (a, c, e, and g) show BSE images of 4 porphyritic basalt fragments. The right four panels (b, d, f, and h) present corresponding Pb-Pb isochrons. Pb isotope analyses were performed on pyroxene, plagioclase, and fine-grained matrix for porphyritic basalt fragments. The results of individual fragments define an imprecise isochron, but most analyses fall onto the ca. 2.8 Ga isochron. Spots that deviate from the isochron are shown as grey circles and are excluded from the regression of the leftmost isochrons. Error bars represent 2 s.e. (standard error). The uncertainties of isochron ages are reported at the 95% confidence (conf.) level. Pl, plagioclase; Pyx, pyroxene; Mtr, matrix.

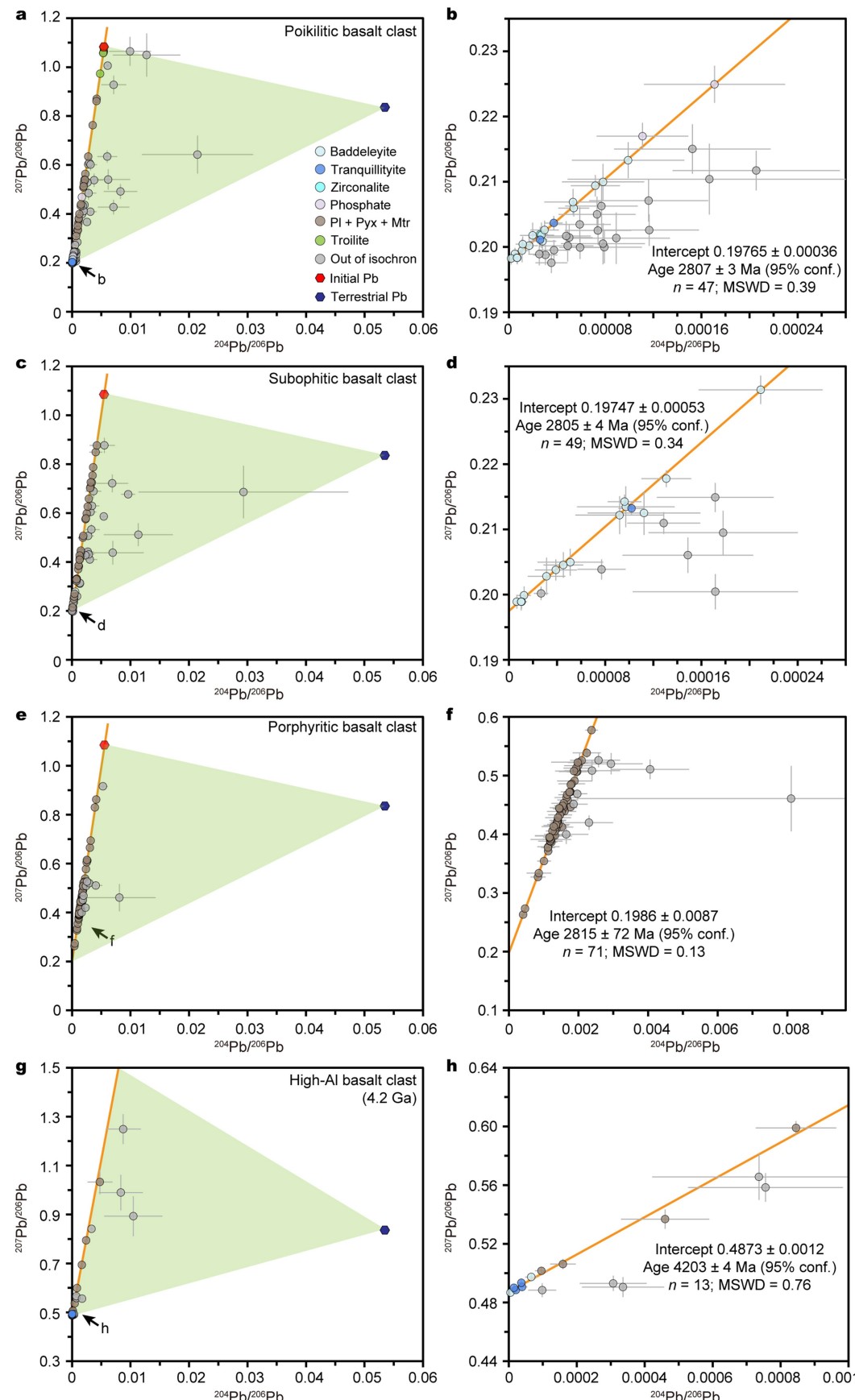

**Extended Data Fig. 5** | See next page for caption.

**Extended Data Fig. 5 | Pb-Pb isochrons for the Chang'e-6 basalts.** The left four plots show the data from the 2.8 Ga basalt fragments with poikilitic (**a**), subophitic (**b**), and porphyritic textures (**c**), as well as from the 4.2 Ga poikilitic fragment (**d**). The right four plots (**b**, **d**, **f**, and **h**) are enlarged sections of the isochrons highlighting the measurements near the *y*-intercepts. The red hexagon represents the lunar initial Pb composition for the 2.8 Ga basalt, while the green triangle areas indicate the mixing trend among radiogenic Pb, initial Pb, and current terrestrial Pb compositions. Spots that deviate from the isochron are shown as grey circles and are excluded from the regression of the leftmost isochrons. Error bars represent 2 s.e. (standard error). The uncertainties of isochron ages are reported at the 95% confidence (conf.) level. MSWD stands for the mean square weighted deviation with no unit. Pl, plagioclase; Pyx, pyroxene; Mtr, matrix.

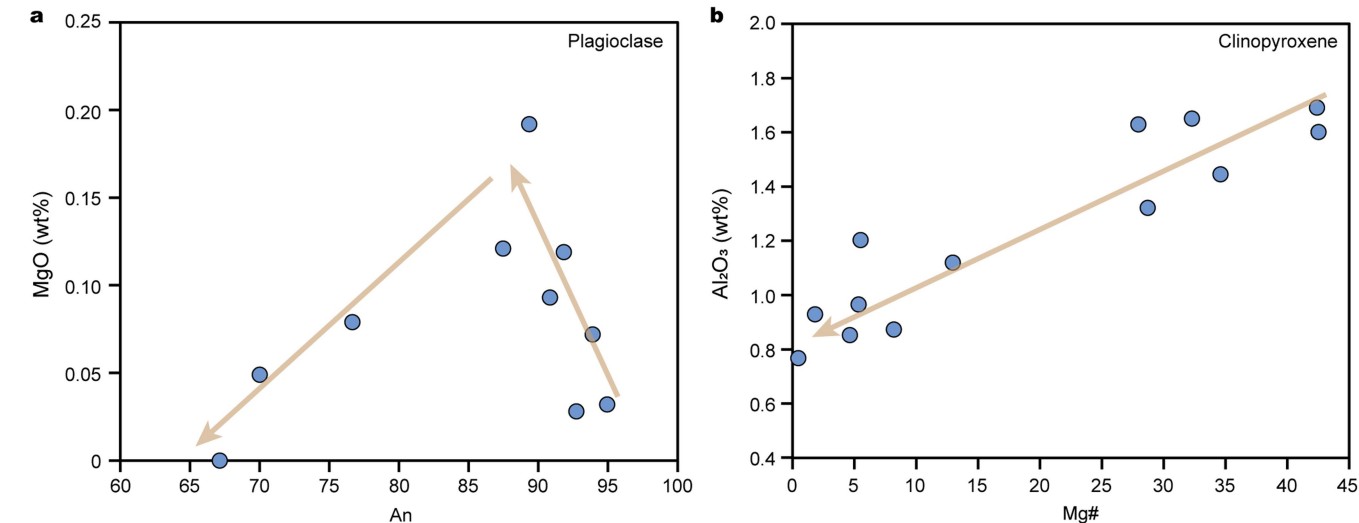

**Extended Data Fig. 6 | Correlations of major element compositions in plagioclase and clinopyroxene from the 4.2 Ga high-Al basalt fragment. a**, Correlation between anorthite content (An) and MgO (wt%) content in plagioclase. **b**, Correlation between Mg# [= molar Mg/(Mg+Fe)] and $Al_2O_3$ (wt%) content in clinopyroxene.

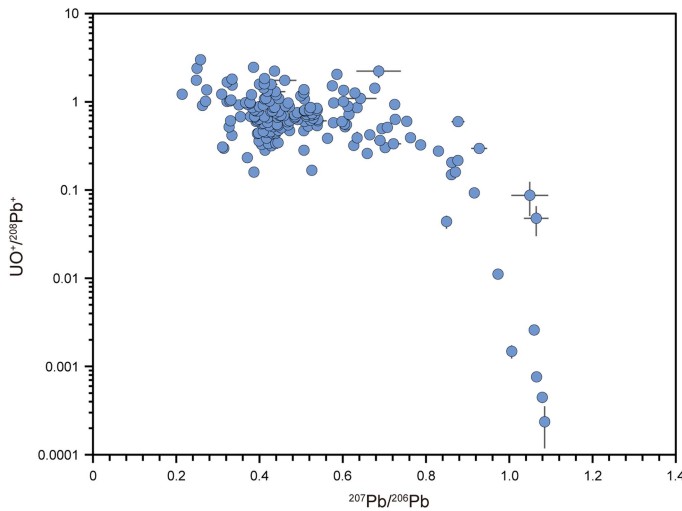

**Extended Data Fig. 7 | UO⁺/²⁰⁸Pb⁺ versus ²⁰⁷Pb/²⁰⁶Pb for points within the analysed rock-forming minerals of the ca. 2.8 Ga Chang'e-6 basalt.** The measured UO⁺/²⁰⁸Pb⁺ ratios are used here to indicate the U/Pb ratios. Error bars represent 1 s.e. (standard error). The point with the lowest UO⁺/²⁰⁸Pb⁺ ratio and the highest ²⁰⁷Pb/²⁰⁶Pb ratio most likely represents the best estimation for the initial Pb composition of the ca. 2.8 Ga Chang'e-6 basalt.

**Extended Data Table 1 | Summary of SIMS Pb isotope analyses for each type of Chang'e-6 basalt fragments**

| | | Porphyritic (2.8 Ga) | Subophitic (2.8 Ga) | Poikilitic (2.8 Ga) | Poikilitic (4.2 Ga) | Total |
|---|---|---|---|---|---|---|
| Basalt clast numbers | | 9 | 45 | 53 | 1 | 108 |
| Zr-bearing minerals grains | Baddeleyite | 0 | 23 | 45 | 2 | 70 |
| | Tranquillityite | 0 | 3 | 4 | 5 | 12 |
| | Zirconolite | 0 | 1 | 2 | 0 | 3 |
| Phosphate grains | | 0 | 4 | 4 | 0 | 5 |
| Other phases | | 79 | 51 | 47 | 16 | 196 |
| Isochron Age (Ma) | | 2,815 ± 72 | 2,805 ± 4 | 2,807 ± 3 | 4,203 ± 4 | - |