## [Peer Review File · Nature]

Lunar farside volcanism 2.8 billion years ago from Chang'e-6 basalts

Corresponding Author: Professor Qiuli Li

Version 1:

Reviewer comments:

Referee #1

(Remarks to the Author)

Overview

This is an incredibly exciting study, and absolutely merits publication in a high-profile journal. It is the first geochronology study to come from the Chang'e-6 samples, and will be of immense importance to the lunar and planetary science community, as well as being of interest to a broader audience. As such, I recommend publication in Nature. However, precisely because the importance of this work, I feel it is crucial to make sure it is as robust and thorough as possible. With this in mind I have made a number of comments and suggestions, which should be incorporated before publication.

Main comments

Discussion of age determinations

My biggest issue with the manuscript is the way the analyses from different rock fragments have been combined into single isochrons without any real discussion of how appropriate this may or may not be. I understand the authors have grouped the data points based on different textural types. However, given how small the individual fragments are, there is no guarantee that the texture of a <1 mm rock fragment is at all representative of the rock from which it was derived. Even if you had larger samples, it would still be questionable to generate an isochron based on (in some cases) several different samples each with just one or two analyses in each.

Having looked through the dataset and re-made the isochrons myself, I do not think this is a fundamental reason to reject the study – indeed, I think the data seem very robust and hold together well. I just think the logic for how it is all presented needs to be modified. My advice would be as follows:

- take each textural type and generate an isochron for each fragment for which the authors have more than four analyses (the poikilitic group is probably the worst as I think there are only three fragments with four or more analyses – but there is still enough here). You do not need to present each individual isochron in the main paper (although they could be included in the supplement), just summarise them (e.g. “for the poikilitic fragments, there were 3 samples for which isochrons were defined, indicating dates of ***Ma...”).
- having done this, you can then plot the analyses from the other fragments of the same textural type on to these isochrons and argue that they are consistent with derivation from the same parent rock (e.g. “the Pb isotopic compositions obtained from the different poikilitic fragments are consistent with derivation from a single parent rock, and can be used to generate a combined isochron, indicating a date of ***Ma...”)
- then you can argue that all three textural types can also be consistent with derivation from the same parent melt and (as you have already done) generate your single combined isochron and age comprising all the different textural types (as you have in Fig. 2)

Note – none of this will change the final ages or even the figures in the manuscript (I have double-checked them myself and get the same results). It is just a question of modifying the text describing how the ages are defined. How much of this you

include in the main manuscript text is debatable. It may push the text beyond the Nature word limit. But it should be possible to summarise this and incorporate the details into the supplementary text.

Discussion of mantle sources

My next biggest criticism relates to the discussion of the mantle sources. In particular, I do not understand the justification for invoking a giant impact event as the only explanation for the mantle source of the high-Al basalt (Line 144-145). Yes, it probably is plausible, but are there other potential mechanisms? Could overturn of mantle cumulates (as is predicted in many LMO evolution models) deliver enough KREEP material to the mantle sources? What about assimilation of KREEP and plagioclase rich material as the magmas rise to the lunar surface? It feels like the SPA impact basin connection has been forced in without any consideration of other possibilities.

On the subject of the mantle source discussion, it would be useful to try and estimate the percentage of KREEP that would be required to produce the source μ -values calculated for these basalts. The authors have already included the references that I would suggest for likely endmember compositions (KREEP endmember from Apollo 15 KREEP basalt 15386 – Snape et al. 2016; primitive mafic cumulate composition from YAMM group meteorite basalts – Merle et al. 2024). I would guess that the 360 μ , 2.8 Ga basalt source would involve less than 1% KREEP. The older 1620 μ basalt source I am less sure about, but it may not be much more than 10-20%. Either way, it would be interesting to check this.

Methodology documentation

My remaining main comments refer to the documentation of the methods and datasets in the supplementary materials.

Ideally, I would also like to see a supplementary file with SEM images of each of the analysed grains. I realise this will be quite a lot of work to assemble and result in a large file, but I think it is worth the effort so that readers can assess the textural similarities and differences between the grains. Currently the authors only provide a limited number of SEM images for what they refer to as 'representative' examples of the textures, plus their petrographic descriptions

For the EMPA analyses, it would be helpful to include a summary table of the reference materials used for the instrument calibration for each element. In general, the EMPA data look good, although I did notice one plagioclase point with a low total ('CE6C0100YJFM002-014-PI-8').

The SIMS data included in the supplementary table is quite minimal, with several values missing, which were apparently measured. For example, the authors state that the second SIMS session included measurements of 208Pb. While they are not as commonly used as the 204Pb, 206Pb and 207Pb values in these kinds of studies, they are potentially still of interest and should be included. Additionally, I like the way the authors measured 238UO₊ and used these to assess the presence of initial Pb compositions – it would be nice to include these values in the supplementary table.

The authors highlight the use of the NIST SRM610 glass reference material for both sets of SIMS measurements. It would be helpful if the measurements of these could be summarised in a separate supplementary table. On the subject of the glass reference material measurements – were the NIST SRM610 measurements used to correct for relative differences in gains for the individual electron multipliers? If so this needs to be stated. If not, I recommend the authors do apply such a correction. While the effect is usually small, it is sometimes significant, especially when dealing with overall uncertainties of 10 million years or less.

The authors state that regular measurements were made of any detector background noise. This is good to know, but they do not provide any indication of the results of these measurements. Some documentation of these measurements should be included in supplementary files. It does not need to be every single background measurement – I would typically include an average of the background levels on each detector, standard deviation and an indication of the number of measurements. Finally, with regard to the calculations in the study.

The authors need to make sure their methodology states the assumptions associated with the calculation of the isochron dates. In particular, they should state the decay constants and the 238U/235U ratio used.

For the calculation of the source μ -values, I would suggest the authors are also a little more explicit here with regard to the model assumptions, rather than simply citing Snape et al. and Li et al. It does not need much, just a simple statement of the main parts of the model – e.g. model starting composition and a reference (Canyon Diablo Troilite – Göpel et al. 1985?), model starting time (4567 Ma? 4500 Ma?), bulk silicate Moon μ -value and the time of differentiation when the separate silicate reservoirs were formed.

Minor comments

Throughout the manuscript the word 'clasts' is used to describe individual rock fragments. To me (and I think most sample scientists), the term 'clast' is used to describe material within a breccia. I would prefer an alternative term – perhaps 'rock fragments' or 'chips'.

Line 22-23 and Line 115-116: The authors state that the 4.2 Ga age for the high-Al basalt is the oldest obtained for any returned lunar sample. This is technically true, but seems a little disingenuous as it ignores evidence of older lunar volcanic

activity found in the lunar meteorite collection: e.g. Kalahari 009 – Ca-phosphate $^{207}\text{Pb}/^{206}\text{Pb}$, Lu-Hf and Sm-Nd dates of ~4.36 Ga (Terada et al. 2007 – <https://doi.org/10.1038/nature06356>; Sokol et al. 2008 – <https://doi.org/10.1016/j.gca.2008.07.012>; Shih et al. 2008 – <https://ui.adsabs.harvard.edu/abs/2008LPI...39.2165S/abstract>; Snape et al. 2018 – <https://doi.org/10.1016/j.epsl.2018.08.035>); MIL 13317 – Pb-Pb isochron dates of basaltic clasts giving an age of 4.33 Ga (Snape et al. 2018).

Line 25: "...mare volcanism yields a diminished source μ value..." – the phrasing here is a bit strange. It is not correct to say that the mare volcanism 'yields' a particular μ value. I would rephrase to something like: "The Pb isotope compositions of these younger basalts indicate derivation from a source with a μ value of ~360."

Line 89: The authors state here that uranium-lead isotopic compositions were determined. But only Pb isotope compositions are presented in the tables (see my comment above regarding the SIMS data table). I would change this to just saying "Pb isotopic compositions were determined".

Line 100: "...with a similar slope..."

Line 106-107: Similar to my comment about the older high-Al basalt, saying 'such an age has never been reported' in this context also feels a bit disingenuous. While technically true, the really important point is that this age provides a crucial intermediate value between the Chang'e-5 and Apollo basalts, which then sets up the final discussion about the crater calibration. So I would rephrase this sentence to make it a little less sensationalist, and a little more informative (e.g. "This age provides an intermediate value between that of the ~2 Ga Chang'e-5 basalts and the ~3.2-3.8 Ga Apollo basalts").

Line 163-165: This statement is broadly okay, but it is not quite true to suggest that the farside mantle sources have a similar range to those of the nearside basalts. The Apollo KREEP basalts are predicted to have been derived from sources with μ values approximately double of those suggested here for the Chang'e-6 high-Al basalt.

Fig. 4: The authors should state whether the error bars represent 2 sigma uncertainties (I assume this is the case, as it is in all the other figures)

Table S1: The authors should state whether the uncertainties provided for each analysis are at the 1 or 2 sigma level (I think it is 1 sigma)

In general the written English in the manuscript is okay. However, it might benefit from some careful proof-reading, ideally by a native English speaker.

Referee #2

(Remarks to the Author)

This paper presents exciting results of the first geochronology analyses of basalts from the farside of the Moon. I want to complement the team on the high quality of data, which demonstrates the meticulous care that was taken during SIMS analyses. The results of the geochronology and initial-Pb modeling are important and are of the caliber of a Nature publication. However, the conclusions that are drawn from these results are not fully justified. Therefore, the manuscript needs moderate to major revisions prior to publication. Below are recommendations for how the discussion and conclusions of the paper could be improved.

1. The identification of a high-Al basalt with a ~4.2 Ga crystallization age and an enriched, KREEP-like initial Pb composition is significant. Finding a KREEP-like enriched signature on the near-side could imply a more global KREEP layer, although this would need to be confirmed with trace element analyses. However, the conclusion that this sample comes from a nearby cryptomare is not well justified.

a. The crater counting age of this region is 4.02 Ga, but the Pb-Pb age of the sample is ~4.2 Ga. This age difference is ~200 Myr, which is likely larger than the uncertainties on the crater counting age. As such, the ages are not in agreement and this alone is not sufficient for arguing that the cryptomare is the source region. Is there any chemical evidence for a more enriched source in the cryptomare region (e.g., a Th-anomaly)?

b. Also, there is a possibility, albeit small, that the same could be from the near side. Basalt samples, including high-Al basalts, have been identified in the regolith coarse fines from Apollo 16, which is far from any mare deposit. There is other evidence of long-distance transport of material on the Moon, such as rays from Tycho and associated antipodal deposits. As such, this possibility needs to be discussed or at least a reasoning for ruling this out needs to be provided.

(Zeigler, R.A., Korotev, R.L., Haskin, L.A., Jollif, B.L. and Gillis, J.J., 2006. Petrography and geochemistry of five new Apollo 16 mare basalts and evidence for post-basin deposition of basaltic material at the site. *Meteoritics & Planetary Science*, 41(2), pp.263-284.)

2. The paper makes the general conclusion that the nearside crater curves work for the farside, which is important. However, the argument for this is not robust and the conclusions can be more specific about which curve is the most ideal.

Furthermore, in the discussion of an anchor point for cratering chronology, it is unclear if the model being presented represents a new model or confirming a previous model. In lines 191-192, there are four age predictions for the Chang'e-6 landing site and the manuscript then states that the sample ages agree with the median value of these ages. However, these estimates vary by 670 Myr, which is a large range in ages, so it should be possible to state something more about which (if

any) model is the best. Figure 4 only presents one model and the data in the supplemental table for this figure lists data points from multiple sources. It is unclear to me if the curve in Figure 4 is a new curve or an existing model. This figure should include the curves for all models and highlight the 670 Myr range in potential ages. There should also be a discussion about how well the new Chang'e 6 data agrees with the updated models from Chang'e 5. Lastly, there is no discussion about potential latitudinal variability in impact rate and resulting uncertainties in the crater counting curves near the poles (e.g., Gallant, J., Gladman, B., & Ćuk, M. (2009). Current bombardment of the Earth–Moon system: Emphasis on cratering asymmetries. *Icarus*, 202(2), 371–382.)

3. If there is space in the manuscript, please add more details about the landing site. At a minimum, additional references for the landing site geology needs to be added. An overview figure of the region in the main or supplemental materials would be helpful, even if this figure is reproduced from a previous paper. I've added a comment in the line-by-line comments to suggest where this information and references should be.

Suggestions for Figures:

Figure 1: This figure is referenced in the text when discussing apatite intragranular phase relation with inclusions. Please add an arrow or some other indicator of this in the figure and note it in the figure caption so it is easier for the reader to find. The Extended Data Fig 1 serves better of this, so please add a reference to Extended Data Fig 1 in the figure caption.

Figure 2: I suggest using Extended Data Fig 2 for Figure 2. The text discusses terrestrial-Pb mixing (or lack thereof) in the main text, so it would be helpful to see this in Figure 2. The team took great care with these samples to reduce terrestrial Pb and it should be showcased here. If the same panel format is used, please put a box in panel A indicating what is shown in panel B. Also, please add a definition of to the figure captions since this figure is referred to in the text before this is defined.

Fig 3: I really like this summary figure and the associated data in the supplementary materials. Having the same color scheme between the background and data points is confusing. At first glance, I thought the red and blue data points were denoting KREEP-rich and KREEP-poor. I suggest choosing a different color scheme for the data points. Also, in the supplementary spreadsheet, please a column for references so that each line/data point in the sheet has a reference. This will make the table more useful to the community.

Fig 4: Please add the four curves that produce the four age estimates in lines 191 – 192. Please label Chang'e 5 and use different symbols for Apollo and Lunar data. Also, make clear in the figure caption if the curve presented here is a new curve based on the data listed in the associated supplementary table, or if it is a previous curve.

Supplementary Tables

Table S1 – What are the units for the Pb values? Counts? Signal intensity?

Fig 3 Data – see comment above about Fig 3.

Line-by-Line Comments

20: "Change-6, which landed"

23: Suggested change... "The main volcanic episode of the Chang'e-6 basalt yielded a surprisingly young eruption age of $2,807 \pm 3$ Ma, which is not recognized from the nearside of the Moon."

26: take out "in contrast"

27: Please make the connection that the depleted source means a depletion in heat-producing elements.

37 & 38: remove "ref"

48: Add reference to landing site geology summary to help place chronology and samples in context.

59: "comprising of clinopyroxene..."

91: Suggested change... "Given that the minerals have variable grain sizes and Pb-concentrations, ..."

95: Are the 53 poikilitic and 45 subophitic clasts all of the clasts in these sub groups? Or were there others that did not give this age? If it is all the clasts, I suggest changing this to something like: "All 52 poikilitic and 45 subophitic clasts yielded..."

116: Is the argument that the age reported in this work is more robust (reliable) or more precise (smaller error bars)? If the statement is that the age is more precise, this should be quantified...i.e., what is a precise age? 1%, 0.01%, 0.001% uncertainty?

142: Here it says the same value as a contemporary high-Al basalt, but in line 116, it states that the Chang'e 5 basalt is the oldest. This seems contradictory.

146: suggested change .."largest of which is..."

147: Is the crater modeling age 4.02, or 4.2? Reference 24 lists a projection of model age to around 4 Ga for cryptomare within the SPA Basin right below the Chang'e landing site. I am just highlighting this in the line-by-line comments since it is discussed in one of the main points above.

150: Why is there a switch from 4.2 to 4.25 Ga here? If the sample is the constraint on the age, then it should be >4.2 Ga. If the 4.25 Ga comes from one of the references or an argument made therein, it should be explained here.

204: "ref" needs to be replaced.

151-154: This sentence is confusing. Do you mean a progressive increase in composition with decreasing age or increasing age? Does it increase from 3.4 down to 3.0, then drop again?

172: I think this should be "KREEP-rich (red)"

179: I suggest using just the term "age estimates" not "absolute age estimates" when discussing crater counting ages.

Absolute ages come from measuring samples, and these help in constraining crater counting ages. But these crater counting ages are still relative ages – i.e., determined relative to a "known" terrain age assuming a flux of impacts.

184: suggested change ... "2.8 Ga Chang'e-6 basalts filling this critical gap in ."

302: please list grit sizes used for polishing and the thickness of the carbon coat
302: define BSE
303: define EDS here instead of on line 304.
307: examined or identified?
338 & 339 & 343: ref needs to be removed
363: please add reference for terrestrial Pb composition

Version 2:

Reviewer comments:

Referee #1

(Remarks to the Author)

The authors have been incredibly thorough in dealing with my comments. I think the manuscript is looking in very good shape now, and am very happy to recommend publication. I have two very minor remaining comments (see below), but I am happy for them to be overseen by the editor and do not suggest any further review of the paper.

I also owe an apology to the authors – I asked about the corrections made to the SIMS data and managed to overlook the fact that this was in fact stated in the original text. Sorry about that!

Congratulations to the authors for such a nice study on these samples. I'm very excited to see this work published!

Comments:

Methods section: If there is space, it might be worth including a very brief comment on the nature of the terrestrial contamination. Having thought more about it since first reading the manuscript and submitting my original review, it struck me as interesting that the effects of this contamination are seen even in new mounts of lunar material. Admittedly, it is nowhere near as pervasive as one sometimes sees in old Apollo thin sections or meteorite samples, but it does still underscore how susceptible these samples are to contamination (due to the generally low Pb concentrations in lunar Pb). Perhaps just a few words to highlight how susceptible lunar samples are to contamination by terrestrial Pb and suggestions of likely sources of the contamination (I have always assumed sample polishing to be the most likely source of contamination with these kinds of studies, I'd be interested to know what the authors think).

Line 105: What is meant by: "Synthetically, analyses of various mineral phases..."? "Synthetically" seems like an odd choice of word. Perhaps rephrase to something like "Combined, the analyses of various mineral phases..." or "Taken together, the analyses..."

Open Access This Peer Review File is licensed under a Creative Commons Attribution 4.0 International License, which permits use, sharing, adaptation, distribution and reproduction in any medium or format, as long as you give appropriate credit to the original author(s) and the source, provide a link to the Creative Commons license, and indicate if changes were

made.

Response to Reviewers

The reviewers' comments are in black.

Our Responses are in blue, while Revisions are in red.

Line numbers refer to manuscript version *without* tracked changes.

Comments from Editor

Dear Professor Li,

Your manuscript entitled "Lunar farside volcanism 2.8 billion years ago from Chang'e-6 basalts" has now been seen by two referees, whose comments are attached below. While they find your work of significant interest, as indeed do we, they have raised some concerns that will first need to be addressed before we can consider the paper further for publication in *Nature*.

Editorially, your manuscript is already in fairly good shape and we could allow you another 500 words or so in the main text to address the referees' comments. There is also ample room in your Methods section to provide the details requested and you may have up to 10 Extended Data items. But the full set of SEM images requested by referee #1 should be presented in a separate Supplementary Information file (rather than as Extended Data items). Also, if possible, please supply a version of your text in MS Word format.

In the meantime we hope that you find the referees' comments helpful, and please do not hesitate to get in touch if there is anything that you would like to discuss further.

Best regards,

John VanDecar

We appreciate your encouraging evaluation of the scientific value of this work. We prepared a separate supplementary information file for the 108 SEM images of the basalt fragments studied.

Referee #1:

Overview

This is an incredibly exciting study, and absolutely merits publication in a high-profile journal. It is the first geochronology study to come from the Chang'e-6

samples, and will be of immense importance to the lunar and planetary science community, as well as being of interest to a broader audience. As such, I recommend publication in Nature. However, precisely because of the importance of this work, I feel it is crucial to make sure it is as robust and thorough as possible. With this in mind I have made a number of comments and suggestions, which should be incorporated before publication.

We appreciate the reviewer's encouraging evaluation of the scientific value of this work.

Main comments

Discussion of age determinations

My biggest issue with the manuscript is the way the analyses from different rock fragments have been combined into single isochrons without any real discussion of how appropriate this may or may not be. I understand the authors have grouped the data points based on different textural types. However, given how small the individual fragments are, there is no guarantee that the texture of a <1 mm rock fragment is at all representative of the rock from which it was derived. Even if you had larger samples, it would still be questionable to generate an isochron based on (in some cases) several different samples each with just one or two analyses in each.

Having looked through the dataset and re-made the isochrons myself, I do not think this is a fundamental reason to reject the study – indeed, I think the data seem very robust and hold together well. I just think the logic for how it is all presented needs to be modified. My advice would be as follows:

- take each textural type and generate an isochron for each fragment for which the authors have more than four analyses (the poikilitic group is probably the worst as I think there are only three fragments with four or more analyses – but there is still enough here). You do not need to present each individual isochron in the main paper (although they could be included in the supplement), just summarise them (e.g. “for the poikilitic fragments, there were 3 samples for which isochrons were defined, indicating dates of ***Ma...”).
- having done this, you can then plot the analyses from the other fragments of the same textural type on to these isochrons and argue that they are consistent with derivation from the same parent rock (e.g. “the Pb isotopic compositions obtained from the different poikilitic fragments are consistent with derivation from a single parent rock, and can be used to generate a combined isochron, indicating a date of ***Ma...”)
- then you can argue that all three textural types can are also consistent with derivation from the same parent melt and (as you have already done) generate

your single combined isochron and age comprising all the different textural types (as you have in Fig. 2)

Note – none of this will change the final ages or even the figures in the manuscript (I have double-checked them myself and get the same results). It is just a question of modifying the text describing how the ages are defined. How much of this you include in the main manuscript text is debatable. It may push the text beyond the Nature word limit. But it should be possible to summarise this and incorporate the details into the supplementary text.

Revision: Good suggestion. We revised the main text accordingly (lines 100-121) and added two figures (Extended Data Fig. 3 and Extended Data Fig. 4) as follows: Initially, we treated each basalt fragment as an individual sample, with potentially different ages and mantle sources. For the poikilitic fragments, Pb isotope analyses of three fragments yield isochrons with consistent slopes around 162 and ages of $2,811 \pm 44$ Ma (Extended Data Fig. 3a,b), $2,811 \pm 7$ Ma (Extended Data Fig. 3c,d), and $2,762 \pm 36$ Ma (Extended Data Fig. 3e,f). Pb isotope compositions obtained from the other 50 poikilitic fragments also align with the ca. 2.8 Ga isochron, suggesting they may be derived from the same episode of parent magma. Synthetically, analyses of various mineral phases with negligible terrestrial Pb contamination from 53 poikilitic fragments construct a combined leftmost isochron yielding a Pb-Pb age of $2,807 \pm 3$ Ma (95% confidence level, and hereafter except where otherwise noted; Extended Data Fig. 5a,b). We applied a similar procedure to subophitic fragments. A leftmost isochron constructed from 18 analyses on a large subophitic fragment yields an age of 2813 ± 23 Ma (Extended Data Fig. 3g,h) and slope of 162 ± 9 , closely matching that of the poikilitic fragments. Together with the Pb isotope compositions of the other 44 subophitic fragments, a leftmost isochron yields a Pb-Pb age of 2805 ± 4 Ma (Extended Data Fig. 5c,d). For porphyritic basalt fragments lacking visible Zr-bearing minerals, Pb isotope analyses were performed on pyroxene, plagioclase, and the fine-grained matrix. The results of any single fragment define an imprecise isochron, nonetheless they fall onto the aforementioned ca. 2.8 Ga isochron (Extended Data Fig. 4). Overall, Pb isotope analyses ($n = 79$) on 9 porphyritic clasts yield a leftmost isochron Pb-Pb age of $2,815 \pm 72$ Ma (Extended Data Fig. 5e,f) with similar slope compared to the dating results of the poikilitic and subophitic basalt clasts within uncertainties, indicating that they were formed during the same volcanic episode despite the variable petrographic textures. Taken together, a total of 167 analyses of various mineral phases with negligible terrestrial Pb contamination construct an integrated leftmost isochron yielding a Pb-Pb age of $2,807 \pm 3$ Ma (Fig. 2a,b).

Extended Data Fig. 3 | Pb-Pb isochrons for representative poikilitic and subophitic basalt fragments in Chang'e-6 samples. The left four panels (a, c, e, and g) show BSE images of three poikilitic and one subophitic basalt fragment. The right four panels (b, d, f, and h) present corresponding Pb-Pb isochrons with consistent ages of ca. 2.8 Ga. Spots that deviate from the isochron are shown as grey circles and are excluded from the regression of the leftmost isochrons. Error bars represent 2 s.e. (standard error). The uncertainties of isochron ages are reported at the 95% confidence (conf.) level. Pl, plagioclase; Pyx, pyroxene; Mtr, matrix.

Extended Data Fig. 4 | Pb-Pb isochrons for representative porphyritic basalt fragments in Chang'e-6 samples. The left four panels (a, c, e, and g) show BSE images of 4 porphyritic basalt fragments. The right four panels (b, d, f, and h) present corresponding Pb-Pb isochrons. Pb isotope analyses were performed on pyroxene, plagioclase, and fine-grained matrix for porphyritic basalt fragments. The results of individual fragments define an imprecise isochron, but most analyses fall onto the ca. 2.8 Ga isochron. Spots that deviate from the isochron are shown as grey circles and are excluded from the regression of the leftmost isochrons. Error bars represent 2 s.e. (standard error). The uncertainties of isochron ages are reported at the 95% confidence (conf.) level. Pl, plagioclase; Pyx, pyroxene; Mtr, matrix.

Discussion of mantle sources

My next biggest criticism relates to the discussion of the mantle sources. In particular, I do not understand the justification for invoking a giant impact event as the only explanation for the mantle source of the high-Al basalt (Line 144-145). Yes, it probably is plausible, but are there other potential mechanisms? Could overturn of mantle cumulates (as is predicted in many LMO evolution models) deliver enough KREEP material to the mantle sources? What about assimilation of KREEP and plagioclase rich material as the magmas rise to the lunar surface? It feels like the SPA impact basin connection has been forced in without any consideration of other possibilities.

Revision: Good suggestion! We modified this part as follows (lines 172-174):

Several mechanisms may account for this hybrid mantle source, such as mantle overturn or a giant impact^{27,28}, which could transport KREEP material into the mantle, or cause the assimilation of KREEP and plagioclase-rich materials as magma ascends to the lunar surface²⁹.

On the subject of the mantle source discussion, it would be useful to try and estimate the percentage of KREEP that would be required to produce the source μ -values calculated for these basalts. The authors have already included the references that I would suggest for likely endmember compositions (KREEP endmember from Apollo 15 KREEP basalt 15386 – Snape et al. 2016; primitive mafic cumulate composition from YAMM group meteorite basalts – Merle et al. 2024). I would guess that the 360 μ , 2.8 Ga basalt source would involve less than 1% KREEP. The older 1620 μ basalt source I am less sure about, but it may not be much more than 10-20%. Either way, it would be interesting to check this.

Response: This is a good suggestion. We tried to estimate the percentage of KREEP using the KREEP endmember from Apollo 15 KREEP basalt 15386 (~4,000; Snape et al. 2016) and the primitive mafic cumulate composition from the YAMM group meteorite basalts (~90; Merle et al. 2024). The results show that the 2.8 Ga basalt source with μ value of ~360 would involve around 7% KREEP. Taking the Chang'e-5 basalt with a source μ of ~680 as a reference sample as a check on this method, its source would involve ~15% KREEP based on the μ mixture calculation; however, the estimated proportion of KREEP is <0.5% as based on Nd isotopes (Tian et al., 2021), or 1–1.5% as based on geochemical analyses (Zong et al., 2022). This inconsistency may be due to the different behavior of U–Pb and Sm–Nd systems, or the assumed endmembers are not right. This may be an important

issue for future study, but is deemed to be beyond the scope of this manuscript.

Methodology documentation

My remaining main comments refer to the documentation of the methods and datasets in the supplementary materials.

Ideally, I would also like to see a supplementary file with SEM images of each of the analysed grains. I realise this will be quite a lot of work to assemble and result in a large file, but I think it is worth the effort so that readers can assess the textural similarities and differences between the grains. Currently the authors only provide a limited number of SEM images for what they refer to as 'representative' examples of the textures, plus their petrographic descriptions

Revision: Done. We provided a supplementary file (Supplementary Fig. 1) of 18 pages that includes the BSE images of all 108 basalt fragments studied. We labeled the serial number of each fragment according to the list in supplementary data tables. We also added a column of "Position of BSE image" in Supplementary Table 1 (Pb isotope data file) for each fragment. Readers can thus refer to each basalt fragment easily.

For the EMPA analyses, it would be helpful to include a summary table of the reference materials used for the instrument calibration for each element. In general, the EMPA data look good, although I did notice one plagioclase point with a low total ('CE6C0100YJFM002-014-Pl-8').

Revision: Done. We provided the summary table of the reference materials used during EMPA analyses (Supplementary Table 3). This analysis of plagioclase did show a low total. We deleted this unqualified datum, which does not change the trend of the other data.

The SIMS data included in the supplementary table is quite minimal, with several values missing, which were apparently measured. For example, the authors state that the second SIMS session included measurements of ^{208}Pb . While they are not as commonly used as the ^{204}Pb , ^{206}Pb and ^{207}Pb values in these kinds of studies, they are potentially still of interest and should be included. Additionally, I like the way the authors measured $^{238}\text{U}^+$ and used these to assess the presence of initial Pb compositions – it would be nice to include these values in the supplementary table.

Revision: Done. We added $^{208}\text{Pb}^+$ and UO^+ to **Supplementary Table 1**. You are right that $\text{UO}^+ / ^{208}\text{Pb}^+$ is good proxy for points of the initial Pb composition. We provide a plot of $^{207}\text{Pb} / ^{206}\text{Pb}$ vs. $\text{UO}^+ / ^{208}\text{Pb}^+$ for matrix minerals, which is now **Extended Data Fig. 7**.

Extended Data Fig. 7 | $\text{UO}^+ / ^{208}\text{Pb}^+$ versus $^{207}\text{Pb} / ^{206}\text{Pb}$ for points within the analysed rock-forming minerals of the ca. 2.8 Ga Chang'e-6 basalt. The measured $\text{UO}^+ / ^{208}\text{Pb}^+$ ratios are used here to indicate the U/Pb ratios. Error bars represent 1 s.e. (standard error). The point with the lowest $\text{UO}^+ / ^{208}\text{Pb}^+$ ratio and the highest $^{207}\text{Pb} / ^{206}\text{Pb}$ ratio most likely represents the best estimation for the initial Pb composition of the ca. 2.8 Ga Chang'e-6 basalt.

The authors highlight the use of the NIST SRM610 glass reference material for both sets of SIMS measurements. It would be helpful if the measurements of these could be summarised in a separate supplementary table. On the subject of the glass reference material measurements – were the NIST SRM610 measurements used to correct for relative differences in gains for the individual electron multipliers? If so this needs to be stated. If not, I recommend the authors do apply such a correction. While the effect is usually small, it is sometimes significant, especially when dealing with overall uncertainties of 10 million years or less.

Revision: Done. Yes, the NIST SRM610 measurements were used to correct for relative differences in gains for the individual electron multipliers, and also to evaluate the external reproducibility. This is stated in the Methods. We provide the NIST SRM610 data in **Supplementary Table 4**.

The authors state that regular measurements were made of any detector background noise. This is good to know, but they do not provide any indication of

the results of these measurements. Some documentation of these measurements should be included in supplementary files. It does not need to be every single background measurement – I would typically include an average of the background levels on each detector, standard deviation and an indication of the number of measurements.

Revision: Done. We have provided an average of the background levels on each detector, standard deviation, and an indication of the number of measurements in Supplementary Table 5.

Finally, with regard to the calculations in the study.

The authors need to make sure their methodology states the assumptions associated with the calculation of the isochron dates. In particular, they should state the decay constants and the $^{238}\text{U}/^{235}\text{U}$ ratio used.

Revision: Done. We added the parameters used as follows (lines 424-425):
The decay constants of 1.55125×10^{-9} for ^{238}U and 9.8485×10^{-11} for ^{235}U , and the $^{238}\text{U}/^{235}\text{U}$ ratio of 137.818 (ref. ⁵⁸) were used in age calculations.

For the calculation of the source μ -values, I would suggest the authors are also a little more explicit here with regard to the model assumptions, rather than simply citing Snape et al. and Li et al. It does not need much, just a simple statement of the main parts of the model – e.g. model starting composition and a reference (Canyon Diablo Troilite – Göpel et al. 1985?), model starting time (4567 Ma? 4500 Ma?), bulk silicate Moon μ -value and the time of differentiation when the separate silicate reservoirs were formed.

Revision: Done. We added the parameters used as follows (lines 432-436):
The parameters used include the model starting Pb isotope composition of $^{204}\text{Pb}/^{206}\text{Pb} = 9.307$ based on Canyon Diablo Troilite⁵⁹, the model starting time of 4,567 Ma for the Solar System and 4,500 Ma for the Moon's formation, the μ -value of bulk silicate Moon of 462 ± 46 (ref. ¹⁸), and the time of $4,376 \pm 18$ Ma for the differentiation marking the formation of distinct silicate reservoirs¹⁸.

Minor comments

Throughout the manuscript the word 'clasts' is used to describe individual rock fragments. To me (and I think most sample scientists), the term 'clast' is used to

describe material within a breccia. I would prefer an alternative term – perhaps ‘rock fragments’ or ‘chips’.

Revision: Done. We changed all mentions of “clast” to “fragment”.

Line 22-23 and Line 115-116: The authors state that the 4.2 Ga age for the high-Al basalt is the oldest obtained for any returned lunar sample. This is technically true, but seems a little disingenuous as it ignores evidence of older lunar volcanic activity found in the lunar meteorite collection: e.g. Kalahari 009 – Ca-phosphate $^{207}\text{Pb}/^{206}\text{Pb}$, Lu-Hf and Sm-Nd dates of ~ 4.36 Ga (Terada et al. 2007)

– <https://doi.org/10.1038/nature06356>; Sokol et al. 2008

– <https://doi.org/10.1016/j.gca.2008.07.012>; Shih et al. 2008

– <https://ui.adsabs.harvard.edu/abs/2008LPI....39.2165S/abstract>; Snape et al. 2018

– <https://doi.org/10.1016/j.epsl.2018.08.035>); MIL 13317 – Pb-Pb isochron dates of basaltic clasts giving an age of 4.33 Ga (Snape et al. 2018).

Revision: Yes, you are right. Considering there is not enough space in the Abstract part, in revision we cited the lunar meteorite collection information in the Age discussion part as follows (lines 132-135):

To date, this is the oldest returned lunar high-Al basalt sample with a precise age determination with $\sim 0.1\%$ uncertainty, comparable to the nearside 4.3–4.0 Ga high-Al basalt returned by Apollo missions²⁰ but younger than the ca. 4.36 Ga high-Al volcanism documented in lunar meteorite Kalahari 009 (a monomict basaltic breccia).

Line 25: “...mare volcanism yields a diminished source μ value...” – the phrasing here is a bit strange. It is not correct to say that the mare volcanism ‘yields’ a particular μ value. I would rephrase to something like: “The Pb isotope compositions of these younger basalts indicate derivation from a source with a μ value of ~ 360 .”

Revision: Done. We changed as suggested (lines 25-26).

Line 89: The authors state here than uranium-lead isotopic compositions were determined. But only Pb isotope compositions are presented in the tables (see my comment above regarding the SIMS data table). I would change this to just saying “Pb isotopic compositions were determined”.

Revision: Done. We changed as suggested (line 93).

Line 100: "...with a similar slope..."

Revision: We rephased the age determination part according to your suggestions.

Line 106-107: Similar to my comment about the older high-Al basalt, saying 'such an age has never been reported' in this context also feels a bit disingenuous. While technically true, the really important point is that this age provides a crucial intermediate value between the Chang'e-5 and Apollo basalts, which then sets up the final discussion about the crater calibration. So I would rephrase this sentence to make it a little less sensationalist, and a little more informative (e.g. "This age provides a intermediate value between that of the ~2 Ga Chang'e-5 basalts and the ~3.2-3.8 Ga Apollo basalts").

Revision: Done. We changed as suggested (lines 123-124).

Line 163-165: This statement is broadly okay, but it is not quite true to suggest that the farside mantle sources have a similar range to those of the nearside basalts. The Apollo KREEP basalts are predicted to have been derived from sources with μ values approximately double of those suggested here for the Chang'e-6 high-Al basalt.

Response: Agree. The farside mantle sources nearly encompass those of most nearside mantle sources, but are still much lower than those of KREEP and high-Al basalts.

Revision: To address this, we changed as follows (lines 190-192):

The wide range of observed μ values of farside mantle sources identified in this study encompasses nearly most of those of nearside mantle sources^{11,12,18,34} except those of three special KREEP and high-Al basalts, implying comparable hemispheric mantle compositions on either side of the Moon.

Fig. 4: The authors should state whether the error bars represent 2 sigma uncertainties (I assume this is the case, as it is in all the other figures)

Revision: Done. The datasets in Figure 4 include the radiometric ages (i.e., Age) and the cumulative frequencies of craters >1 km in diameter (i.e., N(1)). For the

N(1), the error bar represents 1 σ uncertainty. For the Ages, the error bars represent 95% confidence. This is now stated clearly in the revised figure legend (lines 236-237).

Table S1: The authors should state whether the uncertainties provided for each analysis are at the 1 or 2 sigma level (I think it is 1 sigma)

Revision: Done. You are right that all the uncertainties provided for each analysis are at 1 σ level. We have made that clear in **Supplementary Table 1**.

In general the written English in the manuscript is okay. However, it might benefit from some careful proof-reading, ideally by a native English speaker.

Revision: Thanks for your suggestion. We have asked a native English speaker for careful proof reading.

Reference:

- Merle, R.E., Nemchin, A.A., Whitehouse, M.J., Kenny, G.G., Snape, J.F., 2024. Pb isotope signature of a low- μ ($^{238}\text{U}/^{204}\text{Pb}$) lunar mantle component. *Journal of Petrology* 65, egae062.
- Snape, J.F., Nemchin, A.A., Bellucci, J.J., Whitehouse, M.J., Tartèse, R., Barnes, J.J., Anand, M., Crawford, I.A., Joy, K.H., 2016. Lunar basalt chronology, mantle differentiation and implications for determining the age of the Moon. *Earth and Planetary Science Letters* 451, 149–158.
- Tian, H.C., Wang, H., Chen, Y., Yang, W., Zhou, Q., Zhang, C., Lin, H.L., Huang, C., Wu, S.T., Jia, L.H., Xu, L., Zhang, D., Li, X.G., Chang, R., Yang, Y.H., Xie, L.W., Zhang, D.P., Zhang, G.L., Yang, S.H., Wu, F.Y., 2021. Non-KREEP origin for Chang'e-5 basalts in the Procellarum KREEP terrane. *Nature* 600, 59–63.
- Zong, K., Wang, Z., Li, J., He, Q., Li, Y., Becker, H., Zhang, W., Hu, Z., He, T., Cao, K., She, Z., Wu, X., Xiao, L., Liu, Y., 2022. Bulk compositions of the Chang'E-5 lunar soil: Insights into chemical homogeneity, exotic addition, and origin of landing site basalts. *Geochimica et Cosmochimica Acta* 335, 284–296.

Referee #2:

This paper presents exciting results of the first geochronology analyses of basalts from the farside of the Moon. I want to complement the team on the high quality of data, which demonstrates the meticulous care that was taken during SIMS analyses. The results of the geochronology and initial-Pb modeling are important and are of the caliber of a Nature publication. However, the conclusions that are drawn from these results are not fully justified. Therefore, the manuscript needs moderate to major revisions prior to publication. Below are recommendations for how the discussion and conclusions of the paper could be improved.

We appreciate the reviewer's encouraging evaluation of the scientific value of this work, as well as the excellent suggestions to improve the revised manuscript.

1. The identification of a high-Al basalt with a ~ 4.2 Ga crystallization age and an enriched, KREEP-like initial Pb composition is significant. Finding a KREEP-like enriched signature on the near-side could imply a more global KREEP layer, although this would need to be confirmed with trace element analyses. However, the conclusion that this sample comes from a nearby cryptomare is not well justified.
 - a. The crater counting age of this region is 4.02 Ga, but the Pb-Pb age of the sample is ~ 4.2 Ga. This age difference is ~ 200 Myr, which is likely larger than the uncertainties on the crater counting age. As such, the ages are not in agreement and this alone is not sufficient for arguing that the cryptomare is the source region. Is there any chemical evidence for a more enriched source in the cryptomare region (e.g., a Th-anomaly)?

Response: Considering that the crater-counting model ages (~ 3.0 – 2.5 Ga) of the main mare basalt area are consistent with its Pb–Pb age (~ 2.8 Ga) within $\sim 10\%$ uncertainties, the crater-counting age of the cryptomare (~ 4.05 Ga) is likely consistent with the Pb–Pb age of the high-Al basalt (~ 4.2 Ga). We nonetheless agree that the proximity between the modeled age and the Pb–Pb age of the sample is not enough to make this conclusion.

Previous studies have recognized high-Al basalts and identified cryptomare regions in the SPA basin based on remote sensing observations (Kramer et al., 2008, JGR; Ma et al., 2024, Icarus). In addition, Qian et al. (2024) suggests that volcanic materials beneath the cryptomare to the south of the Chang'e-6 landing

site are plagioclase-rich. Therefore, we speculate that the high-Al basalt fragment documented here most likely originated from this nearby cryptomare region. Furthermore, this nearby cryptomare region shows higher Th contents than the Chang'e-6 landing mare basalt unit (Guo et al. 2024, EPSL; shown below), which may be contributed from underlying high-Al basalt with high μ values. Considering that only larger craters can excavate it, this may be the reason why there are relatively few fragments of high-Al basalt in the returned soils.

[REDACTION]

Figure 2c of Guo et al. 2024, EPSL.

- b. Also, there is a possibility, albeit small, that the same could be from the near side. Basalt samples, including high-Al basalts, have been identified in the regolith coarse fines from Apollo 16, which is far from any mare deposit. There is other evidence of long-distance transport of material on the Moon, such as rays from Tycho and associated antipodal deposits. As such, this possibility needs to be discussed or at least a reasoning for ruling this out needs to be provided.
- c. (Zeigler, R.A., Korotev, R.L., Haskin, L.A., Jollif, B.L. and Gillis, J.J., 2006. Petrography and geochemistry of five new Apollo 16 mare basalts and evidence for post-basin deposition of basaltic material at the site. *Meteoritics & Planetary Science*, 41(2), pp.263-284.)

Response: We agree. It is fair to mention this possibility, albeit small. In addition, among the 12 research groups now working with return Chang'e-6 samples, some have already found the same high-Al basalt fragments; but these results have not yet been published.

Revision:

We have added this possibility in the revision (lines 135-143):

Spectral data reveal wide distributions of high-Al basalt on both sides of the Moon^{21,22}, implying a potential nearside ejection origin. However, given that the studied high-Al basalt fragment exhibits a pristine magmatic texture with no evidence of impact-induced modification, we regard it as a local product of the lunar farside rather than ejecta from the nearside. Notably, the crater-counting age of the cryptomare region on the south of Chang'e-6 landing site^{23,24} is approximately 4.05 Ga (ref. ⁴), consistent with the age of the high-Al basalt within uncertainty. Combined with remote sensing observations showing that volcanic materials beneath the cryptomare are plagioclase-rich, the high-Al basalt fragment documented here most likely originated from this nearby cryptomare region.

2. The paper makes the general conclusion that the nearside crater curves work for the farside, which is important. However, the argument for this is not robust and the conclusions can be more specific about which curve is the most ideal. Furthermore, in the discussion of an anchor point for cratering chronology, it is unclear if the model being presented represents a new model or confirming a previous model.

Response: Thank you for the honest comment and request for more clarity. In our previous work on the Chang'e-5 basalts, we found that the Neukum (1983) model was most consistent with the Chang'e-5 radiometric age, and this model is also the most widely used in lunar crater-counting chronology. Yue et al. (2022) updated the Neukum model with the input of Chang'e-5 calibration point. So, in the revised manuscript version, we compared the recently published crater counting results based on those two functions with the radioisotope age of the Chang'e-6 local basalt. The comparison indicates that the cratering chronology function established for the nearside of the Moon is also applicable to the lunar farside. Also, the radioisotope age of 2.8 Ga of the Chang'e-6 local basalt aligns with the median range of published crater-counting model ages within 10% uncertainties using the model of Yue et al. (2022).

Revision:

We have revised Figure 4 and text as follows (lines 221-223):

The eruption age of the local basaltic volcanism at the Chang'e-6 landing site is dated at ca. 2.8 Ga, which aligns with the median range of published crater-counting model ages within 10% uncertainties using the model of Yue et al.⁴⁶.

Fig. 4 | Lunar crater-counting chronology compared to the critical reference point of the radioisotope age of the Chang'e-6 local basalt. The yellow, grey, and cyan circles represent calibration points respectively derived from Apollo, Luna, and Chang'e-5 samples^{41,46,50}. The purple and light blue curves are the previous crater-counting chronology function of ref. ⁴¹ and the updated function incorporating the radioisotope age of the Chang'e-5 basalt^{12,46}, respectively. The red circle represents the radioisotope age of the main volcanic phase of the returned Chang'e-6 samples with the red vertical line showing the range of published $N(1)$ values of the cumulative frequencies of craters >1 km in diameter of the Chang'e-6 landing region indicated by green points^{4,47-49}. The error bars of $N(1)$ values represent 1σ uncertainty, whereas the error bars of age represent 95% confidence.

In lines 191-192, there are four age predictions for the Chang'e-6 landing site and the manuscript then states that the sample ages agree with the median value of these ages. However, these estimates vary by 670 Myr, which is a large range in ages, so it should be possible to state something more about which (if any) model is the best. Figure 4 only presents one model and the data in the supplemental table for this figure lists data points from multiple sources. It is unclear to me if the curve in Figure 4 is a new curve or an existing model. This figure should include the curves for all models and highlight the 670 Myr range in potential ages. There should also be a discussion about how well the new Chang'e 6 data agrees with the updated models from Chang'e 5.

Response: The crater-counting dating method is influenced by many factors, especially by the counting areas, along with the crater production function, chronology function, crater counting method, and even the images used. A detailed

discussion of the uncertainty of crater dating results is beyond the scope of this manuscript. In principle, the radiometric dating result is much more reliable, and such calibration points represent the ground-truthing of the crater-counting method. Thus, any current models require updating thereafter.

Although there are several other models as we have considered in our previous paper on the geochronology of Chang'e-5 (Li et al., 2021), they are inconsistent with the Chang'e-5 basalt age and are seldomly used in lunar surface dating studies. As for the 4 age predictions for the Chang'e-6 surface age, three of the papers use the Yue et al. (2022) model, while the other uses the Neukum (1983) model.

Revision: We have updated **Figure 4** in the revision to compare the radiometric age of Chang'e-6 basalt and the model ages constrained by these two cratering chronology curves (Please see last response). We revised the text as follows (lines 218-223):

The Chang'e-6 landing site is in the medium-Ti region, and the suggested crater-counting ages are ca. 2.40 Ga (ref. ⁴⁷) using the model of Neukum⁴¹, and ca. 2.49 Ga (ref. ⁴⁸), ca. 2.50 Ga (ref. ⁴⁹), and ca. 3.07 Ga (ref. ⁴) using the recently updated model of Yue et al.⁴⁶ with the calibration of the Chang'e-5 anchor point. The eruption age of the local basaltic volcanism at the Chang'e-6 landing site is dated at ca. 2.8 Ga, which aligns with the median range of published crater-counting model ages within 10% uncertainties using the model of Yue et al.⁴⁶.

Lastly, there is no discussion about potential latitudinal variability in impact rate and resulting uncertainties in the crater counting curves near the poles (e.g., Gallant, J., Gladman, B., & Ćuk, M. (2009). Current bombardment of the Earth–Moon system: Emphasis on cratering asymmetries. *Icarus*, 202(2), 371–382.)

Response: Thanks for your suggestions. The cratering asymmetry across the lunar surface is a controversial topic and it is the subject of much research, including the latitudinal variability as mentioned by the reviewer. However, detailed analysis is out of scope of the current manuscript. In our opinion, latitudinal variability does not appear very evident. Both Chang'e-5 and Chang'e-6 landed at higher latitudes than the Apollo missions, nonetheless the current crater-counting chronology appears to work well for all samples within ~10% uncertainties.

3. If there is space in the manuscript, please add more details about the landing site. At a minimum, additional references for the landing site geology needs to be

added. An overview figure of the region in the main or supplemental materials would be helpful, even if this figure is reproduced from a previous paper. I've added a comment in the line-by-line comments to suggest where this information and references should be.

Revision: Done. We have added an overview figure of the region as **Extended Data Figure 1** to show the basalt and cryptomare region in the SPA basin. We cited the recent paper by Li et al. (2024, NSR), which is the first comprehensive paper to describe the CE6 landing site geology based on the returned samples.

Extended Data Fig. 1 | Mare and cryptomare distribution within and surrounding the SPA basin. The cryptomare boundaries are sourced from ref. ²³ and the mare boundaries are from ref. ⁶. The two yellow ellipses are the inner and outer rings of the SPA basin¹⁶, while the red circle is the Apollo crater rim.

Suggestions for Figures:

Figure 1: This figure is referenced in the text when discussing apatite intragranular phase relation with inclusions. Please add an arrow or some other indicator of this in the figure and note it in the figure caption so it is easier for the reader to find. The Extended Data Fig 1 serves better of this, so please add a reference to Extended Data Fig 1 in the figure caption.

Revision: Done. We added color-coded arrows for fine-grained minerals, and also cited a reference to Extended Data Figure 2.

Yellow arrows are used to indicate the location of fine-grained minerals. Four other representative basalt fragments are presented in Extended Data Fig. 2. BSE images of all 108 dated basalt fragments can be found in Supplementary Fig. 1.

Figure 2: I suggest using Extended Data Fig 2 for Figure 2. The text discusses terrestrial-Pb mixing (or lack thereof) in the main text, so it would be helpful to see this in Figure 2. The team took great care with these samples to reduce terrestrial Pb and it should be showcased here. If the same panel format is used, please put a box in panel A indicating what is shown in panel B. Also, please add a definition of μ to the figure captions since this figure is referred to in the text before this is defined.

Response: This is a good suggestion. However, Extended Data Fig. 2 is large and covers a whole page. Figure 2 is simpler and is readily understood by a broad audience; meanwhile, Extended Data Figure 2 is suitable for expert readers. The coordinate range of panel b is just a very small “point” in the panel a, so we cannot add a box to it.

Revision: We have added a definition of μ in the figure captions.

Fig 3: I really like this summary figure and the associated data in the supplementary materials. Having the same color scheme between the background and data points is confusing. At first glance, I thought the red and blue data points

were denoting KREEP-rich and KREEP-poor. I suggest choosing a different color scheme for the data points. Also, in the supplementary spreadsheet, please add a column for references so that each line/data point in the sheet has a reference. This will make the table more useful to the community.

Revision: Good suggestion. We changed the colour scheme for the data points and background for Figure 3. We added a column for references to each data point in the Source Data Figure 3.

Fig. 3 | Mantle source μ values over time for returned nearside and farside lunar basalts, and lunar basalt meteorites. The μ value of the lunar magma ocean (LMO) representing the bulk Moon is indicated by the light grey band. The background colour gradient is associated with KREE-poor (indigo blue) and KREEP-rich (orange) mantle sources as indicated by the corresponding μ values. Data are sourced from refs. ^{11,12,18,34-36,38,39}. Error bars represent 2 s.e. A11, Apollo 11 high-Ti basalts; A12, Apollo 12 low-Ti basalts; A15, Apollo low-Ti basalts; A17, Apollo 17 high-Ti basalts; A14 high-Al, Apollo 14 high-Al basalts; A15 KREEP, Apollo 15 KREEP basalts; CE-5, Chang’e-5 low-Ti basalts; CE-6 high-Al, Chang’e-6 high-Al basalts of ca. 4.2 Ga; CE-6 local, Chang’e-6 local low-Ti basalts of ca. 2.8 Ga.

Fig 4: Please add the four curves that produce the four age estimates in lines 191 – 192. Please label Chang’e 5 and use different symbols for Apollo and Lunar data. Also, make clear in the figure caption if the curve presented here is a new curve based on the data listed in the associated supplementary table, or if it is a previous curve.

Revision: We modified this figure and caption accordingly (shown earlier above). The 4 age estimates are based on two curves, i.e. Neukum et al. (1983) and Yue et al. (2022). What we plot are the previous curves, not a new curve—allowing for an independent comparison of the previous model curve and our new radiometric age.

Supplementary Tables

Table S1 – What are the units for the Pb values? Counts? Signal intensity?

Revision: The unit for Pb intensity is counts per second (cps), which has been added to **Supplementary Table 1**.

Fig 3 Data – see comment above about Fig 3.

Revision: We added a column for references to each data point in the **Source Data Figure 3**.

Line-by-Line Comments

20: “Change-6, which landed”

Revision: Done.

23: Suggested change... “The main volcanic episode of the Chang’e-6 basalt yielded a surprisingly young eruption age of $2,807 \pm 3$ Ma, which is not recognized from the nearside of the Moon.”

Revision: Done.

26: take out “in contrast”

Revision: Done.

27: Please make the connection that the depleted source means a depletion in heat-producing elements.

Revision: Done. We changed it to (lines 27-28):

“...with a shift to a source depleted in heating-producing elements.”

37 & 38: remove “ref”

Revision: Done.

48: Add reference to landing site geology summary to help place chronology and samples in context.

Revision: Done. Here we cited Li et al. (2024, NSR) paper, which provides detailed information about the landing site.

59: “comprising of clinopyroxene...”

Revision: Done.

91: Suggested change... “Given that the minerals have variable grain sizes and Pb-concentrations, ...”

Revision: Done.

95: Are the 53 poikilitic and 45 subophitic clasts all of the clasts in these sub groups? Or were there others that did not give this age? If it is all the clasts, I suggest changing this to something like: “All 52 poikilitic and 45 subophitic clasts yielded...”

Revision: There are three different texture types: poikilitic, subophitic, and porphyritic. We modified the description here as suggested by Reviewer 1 (lines 100-121).

116: Is the argument that the age reported in this work is more robust (reliable) or more precise (smaller error bars)? If the statement is that the age is more precise, this should be quantified....i.e., what is a precise age? 1%, 0.01%, 0.001% uncertainty?

Response: The high-Al basalts returned by Apollo 14 dated by the Rb–Sr isotopic method yielded ages ranging from 4.317 ± 0.17 Ga to 3.937 ± 0.06 Ga (Hui et al., 2013 EPSL). Considering the uncertainties, our newly returned high-Al basalt (4.203 ± 0.004 Ga) thus appears to be the oldest among returned lunar samples to be precisely dated. We added 0.1% age uncertainty here.

142: Here it says the same value as a contemporary high-Al basalt, but in line 116, it states that the Chang'e 6 basalt is the oldest. This seems contradictory.

Revision: Agree. We rephased here as follows (lines 168-170):

The mantle source of the 4.2 Ga high-Al basalt exhibits a μ value of $1,620 \pm 160$, lower than that of ca. 3.95–3.90 Ga nearside high-Al basalts or ca. 3.88 Ga KREEP basalt¹¹, but significantly higher than those of mare basalts (Fig. 3).

146: suggested change ..“largest of which is...”

Revision: Done.

147: Is the crater modeling age 4.02, or 4.2? Reference 24 lists a projection of model age to around 4 Ga for cryptomare within the SPA Basin right below the Chang'e landing site. I am just highlighting this in the line-by-line comments since it is discussed in one of the main points above.

Response: Sorry, this was a typo. The crater modeling age estimates are 4.05 Ga by Qian et al. (2024) and 3.86 Ga by Zeng et al. (2023).

150: Why is there a switch from 4.2 to 4.25 Ga here? If the sample is the constraint on the age, then it should be >4.2 Ga. If the 4.25 Ga comes from one of the references or an argument made therein, it should be explained here.

Revision: You are right. All the estimates for the SPA formation time are >4.2 Ga. Here, correcting to “>4.2 Ga” for SPA formation is enough for context relative to 4.2 Ga high-Al basalt.

204: “ref” needs to be replaced.

Revision: Done. Changed to “Yue et al.”

151-154: This sentence is confusing. Do you mean a progressive increase in composition with decreasing age or increasing age? Does it increase from 3.4 down to 3.0, then drop again?

Revision: You understand well. We have modified here accordingly. There is a progressive increase in μ value from ~300 to ~1,400 with decreasing age from 3.4

to 3.0 Ga; then, μ value drops to ~ 790 for a 2.94 Ga basaltic meteorite and further to ~ 680 for the 2.0 Ga CE5 basalt. We rephased here as follows (lines 177-179):
The apparent progressive increase in μ values of mantle sources from 300–1,400 with decreasing formation age from 3.4–3.0 Ga as recorded in Apollo low-Ti basalts, as well as low-Ti and very-low-Ti basaltic meteorites^{11,18,34}

172: I think this should be “KREEP-rich (red)”

Revision: Thanks for catching this typo. Fixed.

179: I suggest using just the term “age estimates” not “absolute age estimates” when discussing crater counting ages. Absolute ages come from measuring samples, and these help in constraining crater counting ages. But these crater counting ages are still relative ages – i.e., determined relative to a “known” terrain age assuming a flux of impacts.

Revision: Agree. Crater-counting ages are model ages. We deleted “absolute” here.

184: suggested change ... “2.8 Ga Chang’e-6 basalts filling this critical gap in.”

Revision: Done.

302: please list grit sizes used for polishing and the thickness of the carbon coat

Revision: Done (lines 356-358).

The studied Chang’e-6 basalt fragments were embedded in epoxy mounts and then polished using a grinder with fine diamond pastes (grit sizes of 1 μm and 0.25 μm). The samples were coated with a ~ 8 nm carbon layer for SEM analysis.

302: define BSE

Revision: Done.

303: define EDS here instead of on line 304.

Revision: Done.

307: examined or identified?

Revision: Identified.

338 & 339 & 343: ref needs to be removed

Revision: Done.

363: please add reference for terrestrial Pb composition

Revision: Done. We added the reference Stacey and Kramers. (1975).

References:

- Guo, D., Bao, Y., Liu, Y., Wu, X., Xu, Y., Yang, Y., Zhang, F., Jolliff, B., Li, S., Zhao, Z., Huang, L., Liu, J., Zou, Y., 2024. Geological investigation of the lunar Apollo basin: From surface composition to interior structure. *Earth and Planetary Science Letters* 646, 118986.
- Hui, H., Neal, C.R., Shih, C.-Y., Nyquist, L.E., 2013. Petrogenetic association of the oldest lunar basalts: Combined Rb–Sr isotopic and trace element constraints. *Earth and Planetary Science Letters* 373, 150–159.
- Kramer, G.Y., Jolliff, B.L., Neal, C.R., 2008. Distinguishing high-alumina mare basalts using Clementine UVVIS and Lunar Prospector GRS data: Mare Moscoviense and Mare Nectaris. *Journal of Geophysical Research: Planets* 113.
- Li, C., Hu, H., Yang, M.-F., Liu, J., Zhou, Q., Ren, X., Liu, B., Liu, D., Zeng, X., Zuo, W., Zhang, G., Zhang, H., Yang, S., Wang, Q., Deng, X., Gao, X., Su, Y., Wen, W., Ouyang, Z., 2024. Nature of the lunar farside samples returned by the Chang'E-6 mission. *National Science Review*.
- Li, Q.L., Zhou, Q., Liu, Y., Xiao, Z., Lin, Y., Li, J.H., Ma, H.X., Tang, G.Q., Guo, S., Tang, X., Yuan, J.Y., Li, J., Wu, F.Y., Ouyang, Z., Li, C., Li, X.H., 2021. Two-billion-year-old volcanism on the Moon from Chang'e-5 basalts. *Nature* 600, 54–58.
- Ma, M., Chen, J., Chen, S., Li, B., Han, C., Tian, P., 2024. High alumina basalts identification and their feature analysis in Mare Fecunditatis. *Icarus* 407, 115464.
- Neukum, G., 1983. Meteoritenbombardement und datierung planetarer oberflächen. Habilitation Dissertation for Faculty Membership, Ludwig-Maximilians-Univ.

- Qian, Y., Head, J., Michalski, J., Wang, X., van der Bogert, C.H., Hiesinger, H., Sun, L., Yang, W., Xiao, L., Li, X., Zhao, G., 2024. Long-lasting farside volcanism in the Apollo basin: Chang'e-6 landing site. *Earth and Planetary Science Letters* 637.
- Stacey, J. S. & Kramers, J. D. Approximation of terrestrial lead isotope evolution by a two-stage model. *Earth and Planetary Science Letters* **26**, 207-221 (1975).
- Yue, Z., Di, K., Wan, W., Liu, Z., Gou, S., Liu, B., Peng, M., Wang, Y., Jia, M., Liu, J., Ouyang, Z., 2022. Updated lunar cratering chronology model with the radiometric age of Chang'e-5 samples. *Nature Astronomy* 6, 541–545.
- Zeng, X., Liu, D., Chen, Y., Zhou, Q., Ren, X., Zhang, Z., Yan, W., Chen, W., Wang, Q., Deng, X., Hu, H., Liu, J., Zuo, W., Head, J.W., Li, C., 2023. Landing site of the Chang'e-6 lunar farside sample return mission from the Apollo basin. *Nature Astronomy* 7, 1188–1197.

Response to Reviewers

The reviewers' comments are in black.

Our **Responses** are in blue, while **Revisions** are in red.

Line numbers refer to manuscript version *without* tracked changes.

Referee #1:

The authors have been incredibly thorough in dealing with my comments. I think the manuscript is looking in very good shape now, and am very happy to recommend publication. I have two very minor remaining comments (see below), but I am happy for them to be overseen by the editor and do not suggestion any further review of the paper.

I also owe an apology to the authors – I asked about the corrections made to the SIMS data and managed to overlook the fact that this was in fact stated in the original text. Sorry about that!

Congratulations to the authors for such a nice study on these samples. I'm very excited to see this work published!

Response: We appreciate the reviewer's encouraging evaluation of the scientific value of this work.

Comments:

Methods section: If there is space, it might be worth including a very brief comment on the nature of the terrestrial contamination. Having thought more about it since first reading the manuscript and submitting my original review, it struck me as interesting that the effects of this contamination are seen even in new mounts of lunar material. Admittedly, it is nowhere near as pervasive as one sometimes sees in old Apollo thin sections or meteorite samples, but it does still underscore how susceptible these samples are to contamination (due to the generally low Pb concentrations in lunar Pb). Perhaps just a few words to highlight how susceptible lunar samples are to contamination by terrestrial Pb and suggestions of likely sources of the contamination (I have always assumed sample

polishing to be the most likely source of contamination with these kinds of studies, I'd be interested to know what the authors think).

Response: I agree with you that sample polishing should be an important source of contamination with these kinds of studies, especially when the epoxy mount has low hardness. The susceptible positions are grain boundaries and cracks, where the coating materials (carbon or gold), even polishing materials could be sunk. It is better to use carbon coating rather than gold coating to decrease the possible contamination.

Revision: We added a sentence to highlight this point in 'Methods':

The likely sources of the terrestrial Pb contamination are the coating materials (with carbon being preferable over gold for reducing terrestrial Pb contamination) and residual polishing materials, which could sink into grain boundaries and cracks.

Line 105: What is meant by: "Synthetically, analyses of various mineral phases...?"

"Synthetically" seems like an odd choice of word. Perhaps rephrase to something like "Combined, the analyses of various mineral phases..." or "Taken together, the analyses..."

Revision: Done. we rephrase here to "Taken together, the analyses of"